# IP$_6$-assisted CSN-COP1 competition regulates a CRL4-ETV5 proteolytic checkpoint to safeguard glucose-induced insulin secretion

Hong Lin[1,6], Yuan Yan[1,6], Yifan Luo[1,2,6], Wing Yan So[3,6], Xiayun Wei[1], Xiaozhe Zhang[1], Xiaoli Yang[1], Jun Zhang[1], Yang Su[1], Xiuyan Yang[1], Bobo Zhang[1], Kangjun Zhang[4], Nan Jiang[4], Billy Kwok Chong Chow [2], Weiping Han[3], Fengchao Wang [5] & Feng Rao [1✉]

COP1 and COP9 signalosome (CSN) are the substrate receptor and deneddylase of CRL4 E3 ligase, respectively. How they functionally interact remains unclear. Here, we uncover COP1–CSN antagonism during glucose-induced insulin secretion. Heterozygous $Csn2^{WT/K70E}$ mice with partially disrupted binding of IP$_6$, a CSN cofactor, display congenital hyper-insulinism and insulin resistance. This is due to increased Cul4 neddylation, CRL4$^{COP1}$ E3 assembly, and ubiquitylation of ETV5, an obesity-associated transcriptional suppressor of insulin secretion. Hyperglycemia reciprocally regulates CRL4-CSN versus CRL4$^{COP1}$ assembly to promote ETV5 degradation. Excessive ETV5 degradation is a hallmark of $Csn2^{WT/K70E}$, high-fat diet-treated, and ob/ob mice. The CRL neddylation inhibitor Pevonedistat/ MLN4924 stabilizes ETV5 and remediates the hyperinsulinemia and obesity/diabetes phe-notypes of these mice. These observations were extended to human islets and EndoC-βH1 cells. Thus, a CRL4$^{COP1}$-ETV5 proteolytic checkpoint licensing GSIS is safeguarded by IP$_6$-assisted CSN-COP1 competition. Deregulation of the IP$_6$-CSN-CRL4$^{COP1}$-ETV5 axis underlies hyperinsulinemia and can be intervened to reduce obesity and diabetic risk.

[1] School of Life Sciences, Department of Biology, Southern University of Science and Technology, Shenzhen, Guangdong, China. [2] School of Biological Sciences, The University of Hong Kong, Pokfulam, Hong Kong. [3] Singapore Bioimaging Consortium, Agency for Science, Technology, and Research, Singapore, Singapore. [4] Department of Hepatic Surgery, the Third People's Hospital of Shenzhen and the Second Affiliated Hospital of Southern University of Science and Technology, Shenzhen, Guangdong, China. [5] National Institute of Biological Sciences, Beijing, China. [6] These authors contributed equally: Hong Lin, Yuan Yan, Yifan Luo, Wing Yan So. ✉email: raof@sustech.edu.cn

The cullin-rING ubiquitin ligases (CRLs) are the largest family of E3 enzymes that work together with ubiquitin-activating E1 and ubiquitin-conjugating E2 enzymes to ubiquitylate protein substrates for proteasomal degradation or signal transduction. CRLs are composed of a scaffold Cullin protein (Cul1, 2,-3, 4A/B, 5, or 7), an E2-recruiting RING protein Rbx1/2 (also called Roc1/2), and Cullin-specific adaptors (e.g., DDB1 for CRL4) and/or substrate receptors (e.g., DDB1–CUL4-associated factors (DCAFs) for CRL4)[1,2]. While CRLs ubiquitylate numerous proteins involved in diverse biological processes and are emerging therapeutic targets and tools in cancer research[3,4], their roles in glucose homeostasis and metabolic diseases are less understood[5].

Glucose homeostasis is controlled by circulating insulin levels and insulin signaling. Insulin levels, in particular, is under tight regulation, with hypersecretion/hyposecretion and dysregulated clearance all posing diabetes risk[5]. While several mediators of insulin signaling are CRL substrates[6,7], direct involvement of CRLs in insulin secretion has not been reported. The COP1 protein, first identified in *Arabidopsis* based on the constitutive photomorphogenesis (COP) phenotype of its mutants, is a candidate CRL4 substrate receptor that promotes murine insulin secretion by ubiquitylating the ETV family of transcription factors[8,9], among which, ETV5 is strongly associated with obesity and T2D according to genome-wide association studies (GWAS)[10–12]. Such GWAS link often relates to islet function[13]. However, it remains unknown if any physiologic stimuli regulate insulin release via COP1-mediated ETV degradation. Moreover, since COP1 contains a RING domain and can serve as an E3 ligase independent of CRL4, whether CRL4 is required for COP1's function in insulin secretion remains to be validated.

The E3 ligase activity of CRL is regulated by cycles of neddylation, an ubiquitin-like Nedd8 modification required for CRL activity[14–16]. The eight-subunit COP9 Signalosome (CSN) complex, another mediator of *Arabidopsis* photomorphogenesis[17], is a deneddylase that catalyzes Cullin deneddylation while also sterically inhibiting substrate access[18–21]. We have identified inositol hexakisphosphate (IP$_6$), an abundant cellular metabolite[22], as a "glue"-like molecule facilitating CRL4–CSN complex assembly[23], and also elucidated the structural mechanisms explaining the biochemical role of IP$_6$ at CRL–CSN interface[24,25]. IP$_6$ primarily binds to a basic pocket in CSN2 centered on Lys70, but also makes direct contact with CRL to bridge CRL–CSN interactions[25]. Based on these mechanistic insights, we generated IP$_6$-binding-deficient CSN2-K70E mutant mice. The early embryonic lethality of homozygous *Csn2$^{K70E/K70E}$* mutants underscores the essentiality of IP$_6$ as a CSN cofactor, but also hampered the elucidation of physiological contexts, whereby CRL is regulated by the IP$_6$ metabolite.

Here, by searching for haploinsufficient phenotypes in the viable heterozygous *Csn2$^{WT/K70E}$* mice, we uncovered a glucose-responsive IP$_6$–CSN–CRL4$^{COP1}$–ETV5 axis in insulin secretion control. The anti-obese/diabetic implications of this regulatory axis are further validated by preclinical testing of the CRL neddylation inhibitor MLN4924 (also known as pevonedistat[26]) in high-fat diet (HFD) and ob/ob mouse disease models. Mechanistically, COP1 is an unusual CRL substrate receptor that directly competes with CSN. IP$_6$ binding to CSN modulates mutually exclusive assembly of the active CRL4$^{COP1}$ E3 ligase and the inert CRL4–CSN complex, with such CRL4 toggling regulated by glycemic status. Our study therefore extends the evolutionarily conserved interaction between COP1 and COP9 from photomorphogenesis in plants to glucose homeostasis in mammals, two fundamental energy coping processes.

## Results

**Heterozygous *Csn2$^{WT/K70E}$* mice with partial abolishment of the IP$_6$ cofactor display congenital hyperinsulinism, and**

**Table 1 Genotypes of offspring from *Csn2$^{WT/K70E}$* intercrosses.**

| Genotyping stage | Total no. of embryos | No. with genotype: | | |
|---|---|---|---|---|
| | | *WT/WT* | *WT/K70E* | *K70E/K70E* |
| Live birth | 166 | 48 | 118 | 0 |
| E6.5/7.5 | 51 | 12 | 39 | 0 |
| E3.5 | 19 | 5 | 10 | 4 |

**uncoupling between glucose and insulin tolerance**. Using a combination of structural and biochemical approaches, we recently showed that IP$_6$ promotes CRL deneddylation by acting as a CSN cofactor via direct interaction with a surface patch on CSN2 centered on Lys70 (ref. [25]). To understand the mammalian physiology of IP$_6$-mediated CRL–CSN complex formation, we attempted to generate a mouse line carrying the CSN2-K70E mutation. The K70E mutation selectively abolishes IP$_6$-augmented CSN–CRL interaction without disrupting CSN2 overall structure or its integration into CSN holo-complex[25], ensuring that potential phenotypes originate specifically from loss of the CSN cofactor IP$_6$. Homozygous *Csn2$^{K70E/K70E}$* embryos die at the stage of peri-implantation (Table 1), similar to the *Csn2* null mice[27]. We reasoned that partial CSN dysfunction better mimics tolerable CSN inhibition[28], and therefore searched for haploinsufficient phenotypes in the viable *Csn2$^{WT/K70E}$* heterozygous mice.

The *Csn2$^{WT/K70E}$* heterozygous mice are born in Mendelian ratio from WT-Het intercross and weigh similarly to wild-type littermates before adulthood, but are significantly heavier afterward (Fig. 1a), despite less food intake (Fig. S1a). This leads us to examine their metabolic parameters. Compared to 2.5-month wild-type littermates, the *Csn2$^{WT/K70E}$* mice have normal levels of fed blood glucose, free fatty acid, and triglyceride (Fig. 1b and Fig. S1b, c), but display fasting hypoglycemia (Fig. 1b), hyperinsulinemia (Fig. 1c), and elevated levels of the degradation-resistant proinsulin cleavage product: C-peptide (Fig. 1d), indicating enhanced insulin secretion. When challenged with tolerance tests, the *Csn2$^{WT/K70E}$* mice are more glucose tolerant (Fig. 1e), but are mildly resistant to insulin (Fig. 1f), with more pronounced insulin resistance at 4-month age (Fig. S1d). This rare uncoupling between glucose tolerance and insulin sensitivity further suggests that hyperinsulinemia is not driven primarily by insulin resistance, which would also cause impaired glucose tolerance and hyperglycemia[5], phenotypes not found in *Csn2$^{WT/K70E}$* mice. Treatment with the insulin sensitizer metformin, following a previous protocol[29], significantly improves insulin sensitivity of *Csn2$^{WT/K70E}$* mice (Fig. S1e), but hyperinsulinemia remains (Fig. S1f), again suggesting that hyperinsulinemia is unlikely to be downstream of insulin resistance in *Csn2$^{WT/K70E}$* mice.

Chronic exposure to excess insulin is known to induce insulin resistance[30–34], we therefore investigated whether insulin oversecretion underlies the observed metabolic derangements in *Csn2$^{WT/K70E}$* mice. Decreasing insulin secretion by the β-cell toxin streptozocin eliminates the differences in insulin sensitivity (Fig. 1g), serum insulin levels (Fig. 1h), and body weight (Fig. S1g) between wild-type and *Csn2$^{WT/K70E}$* mice, suggesting that these differences originate from pancreatic β cells. Importantly, both *Csn2$^{WT/K70E}$* mice (Fig. 1i) and their pancreatic islets (Fig. 1j) secrete more insulin under basal conditions, or when stimulated with glucose or another insulin secretagogues (L-arginine; Fig. 1k). The first and second phases of glucose-stimulated islet insulin secretion are both augmented (Fig. 1j). Together, these data suggest that, due at least in part to enhanced insulin secretion, adult *Csn2$^{WT/K70E}$* mice develop early signs of type II diabetes, including congenital hyperinsulinism, insulin resistance, and obesity, though not yet diabetic.

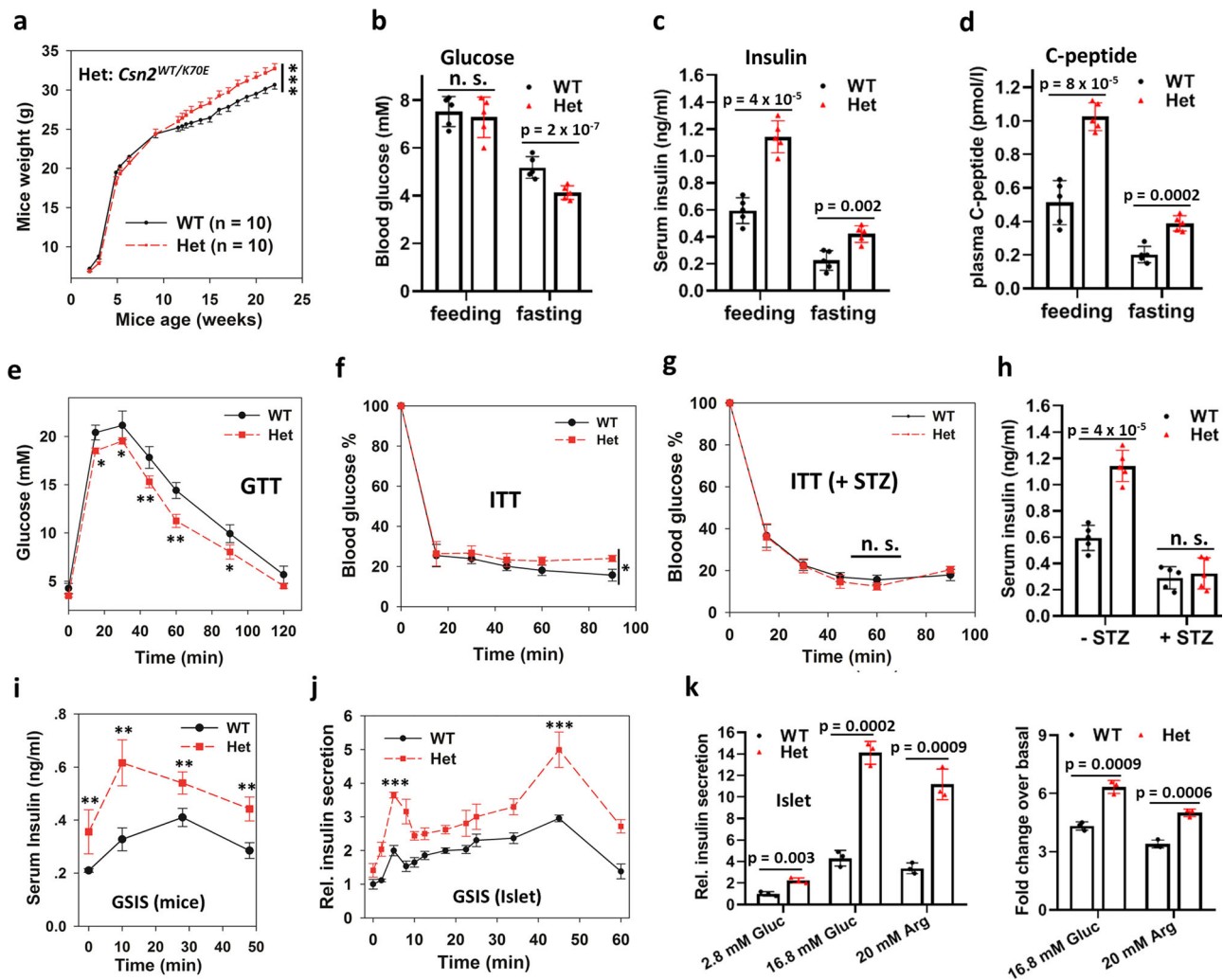

**Fig. 1 Enhanced insulin secretion and congenital hyperinsulinemia in $Csn2^{WT/K70E}$ (Het) mice. a** Body weight measurement of wild-type and $Csn2^{WT/K70E}$ (Het) mice at the indicated age (***$p < 0.001$, calculated by two-way repeated measures ANOVA testing genotype × time effect). Data are presented as mean ± SEM. **b** Feeding and fasting (20 h) glucose levels ($n = 5$). Data are presented as mean ± SD. $P$ values were calculated by two-tailed Student's $t$ test. **c**, **d** Feeding and fasting (16 h) insulin (**c**) and C-peptide (**d**) levels ($n = 5$). Data are presented as mean ± SD. $P$ values were calculated by two-tailed Student's $t$ test. **e** Intraperitoneal glucose tolerance test [$n = 5$ (WT), or 6 (Het), *$p < 0.05$, **$p < 0.01$, calculated by two-tailed Student's $t$ test.]. Data are presented as mean ± SD. **f** Intraperitoneal insulin tolerance test (ITT; $n = 5$ (WT), or 6 (Het), *$p = 0.012$, calculated by two-way repeated measures ANOVA testing genotype × time effect). Data are presented as mean ± SD. 100% blood glucose means 6.37 ± 0.6 mM for WT and 7.27 ± 0.8 mM for Het. **g**, **h** Effect of streptozocin (STZ) treatment (50 mg/kg, 4 days) on serum insulin levels (**g**) and insulin tolerance test (**h**) of wild-type and $Csn2^{WT/K70E}$ (Het) mice, ($n = 5$). Data are presented as mean ± SD. 100% blood glucose means 14.0 ± 5.4 mM for WT and 11.2 ± 2.2 mM for Het. $P$ values were calculated by two-tailed Student's $t$ test. **i** Glucose-stimulated insulin secretion in mice [$n = 4$ (WT), or 5 (Het), **$p < 0.01$, calculated by two-tailed Student's $t$ test.]. Data are presented as mean ± SEM. **j** Glucose-stimulated insulin secretion in isolated pancreatic islets ($n = 3$, ***$p < 0.001$, calculated by two-tailed Student's $t$ test.). Data are presented as mean ± SEM. **k** Basal and secretagogue-stimulated insulin secretion (1 h) in isolated islets ($n = 3$). Data are presented as mean ± SD. Islets from $Csn2^{WT/K70E}$ mice secrete more insulin with L-arginine stimulation, suggesting that altered membrane potential or $Ca^{2+}$ flux are not major causes of their enhanced insulin secretion. $P$ values were calculated by two-tailed Student's $t$ test.

**Insulin production and vesicle release are increased in $Csn2^{WT/K70E}$ mice.** We next examined which molecular events in the islet insulin production and secretion pathway are altered in $Csn2^{WT/K70E}$ mice. Histological, immunohistochemical, and cytoplasmic $Ca^{2+}$ recording experiments showed that islet number (Fig. 2a), size (Fig. 2b), α- and β-cell percentages (Fig. 2c), and glucose- or KCl-induced $Ca^{2+}$ influx (Fig. 2d, e) are comparable between wild-type and $Csn2^{WT/K70E}$ mice. In contrast, total islet insulin content, assayed by immunofluorescence (Fig. 2c) and ELISA (Fig. 2f), are higher in $Csn2^{WT/K70E}$ islets, which is further supported by increased levels of *Ins1* and *Ins2* mRNA (Fig. 2g). Electron microscopic analysis reveal significantly more docked insulin granules in $Csn2^{WT/K70E}$ islets (Fig. 2h), suggesting

augmented vesicle release. Consistently, insulin secretion measured as percentage of total insulin content is also higher in $Csn2^{WT/K70E}$ islets (Fig. 2i). Together, these data suggest that both insulin biosynthesis and insulin vesicle release are augmented in $Csn2^{WT/K70E}$ mice, resulting in hyperinsulinemia.

**CRL4$^{COP1}$-mediated ubiquitylation and degradation relieves the ETV5 transcriptional "brake" to enhance insulin secretion in $Csn2^{WT/K70E}$ mice.** CSN deneddylates and inactivates CRL in an $IP_6$-dependent manner[23,25]. To elucidate the molecular mechanism underlying enhanced insulin secretion in $Csn2^{WT/K70E}$ mice, we first verified that this mutation indeed disrupts CRL–CSN interaction. Consistent with such an expectation,

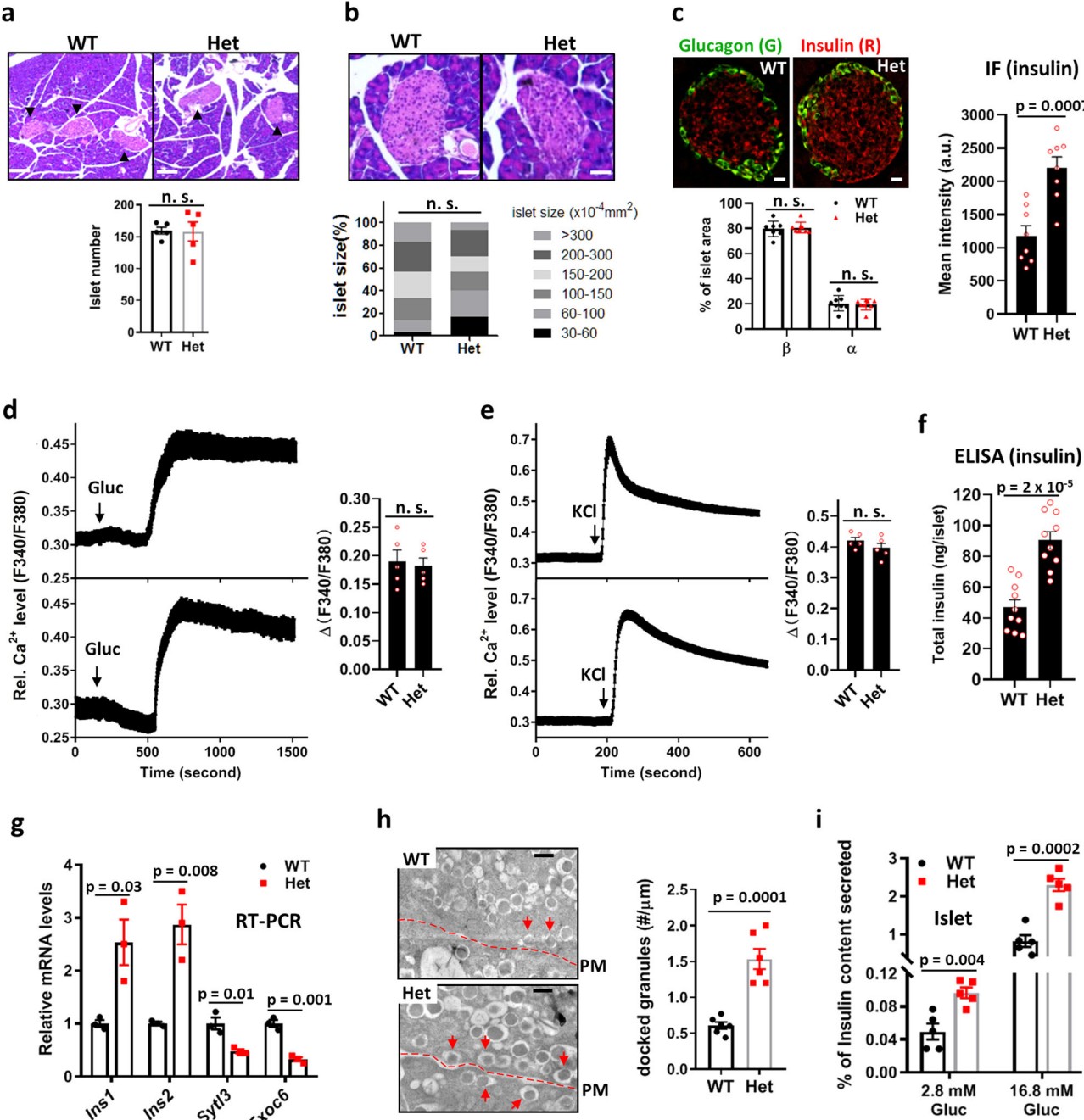

**Fig. 2 Increased insulin production and vesicle release contribute to hyperinsulinemia in *Csn2^WT/K70E* (Het) mice. a, b** H&E staining-based morphometric analysis of pancreatic islets from wild-type and *Csn2^WT/K70E* mice. Number of islets per pancreas (**a**) and average islet area (**b**) was measured by examining 20 slides with every five slides in between, covering the whole pancreas ($n = 5$, n.s. not significant). Data are presented as mean ± SEM. Scare bars, 1.5 μm (**a**) and 0.4 μm (**b**). **c** Immunofluorescence staining of insulin and glucagon in pancreatic islets from wild-type and *Csn2^WT/K70E* mice. Scare bars, 0.2 μm. Lower panel: percentage of α or β cells in an islet is measured based on the area of green (anti-glucagon, α cells) and red fluorescence (anti-insulin, β cells; $n = 8$, n.s. not significant). Data are presented as mean ± SEM. Right panel: quantification of the immunofluorescence intensity of insulin staining ($n = 8$). *P* values were calculated by two-tailed Student's *t* test. **d, e** Average islet cytoplasmic $Ca^{2+}$ responses to depolarization induced by 20 mM glucose (**d**) or 25 mM KCl (**e**), assayed with Fura2AM ($n = 5$, n.s. not significant). Data are presented as mean ± SEM. **f** Total insulin content of pancreatic islet measured using ELISA assay ($n = 10$). *P* values were calculated by two-tailed Student's *t* test. **g** Levels of *Ins1, Ins2, Sytl3,* and *Exoc6* transcripts from pancreas measured by real-time PCR ($n = 10$). Data are presented as mean ± SEM. *P* values were calculated by two-tailed Student's *t* test. **h** Representative electron micrographs of pancreatic islets showing β cell insulin granules. Dotted lines: plasma membrane (PM). Docked granules are marked by arrow heads. Scale bars, 0.5 μm ($n = 6$). Data are presented as mean ± SEM. *P* values were calculated by two-tailed Student's *t* test. **i** GSIS of wild-type and Het islets measured as percentage of insulin secretion. Islets were lysed after GSIS to determine total insulin content ($n = 5$). Data are presented as mean ± SEM. *P* values were calculated by two-tailed Student's *t* test.

CSN3 immunoprecipitation of Cul4 is significantly diminished in $Csn2^{WT/K70E}$ pancreas (Fig. S2a). Furthermore, IP6 addition increases CSN3-Cul4 co-immunoprecipitation in both wild-type and CSN2$^{WT/K70E}$ mice, suggesting that the remaining wild-type allele still responds to the bridging effect of $IP_6$. These data support our proposition that the $Csn2^{WT/K70E}$ mice display partial loss of the $IP_6$ cofactor and consequent CRL regulation.

To validate that enhanced insulin secretion in $Csn2^{WT/K70E}$ mice is due to diminished CRL regulation, we examined the effects of CRL re-inhibition by using the well-characterized CRL neddylation inhibitor: pevonedistate/MLN4924 (ref. [26,35]). While MLN4924 treatment does not affect depolarization-induced $Ca^{2+}$ influx (Fig. S2b), it markedly suppresses basal and glucose-stimulated islet insulin secretion, with more pronounced inhibition under high glucose (Fig. 3a). Weak but significant GSIS is still observed in the presence of MLN4924, suggesting that the islets retain glucose responsiveness to some extent (Fig. 3a). However, MLN4924 eliminates the difference between wild-type and $Csn2^{WT/K70E}$ islets (Fig. 3a), suggesting that CRL disinhibition differentiates wild-type and $Csn2^{WT/K70E}$ islets. When applied in mice, MLN4924 restores serum insulin levels (Fig. 3b) and body weight (Fig. 3c) of the $Csn2^{WT/K70E}$ mice to normal levels over time, without affecting the insulin levels and body weight of wild-type littermates (Fig. 3b, c), nor their food intake (Fig. S2c). Thus, CRL re-inhibition reverses congenital hyperinsulinism and obesity of $Csn2^{WT/K70E}$ mice.

To identify specific CRL(s) that is dysregulated in $Csn2^{WT/K70E}$ mice, we then examined the neddylation status of Cullins. Among all the canonical Cullins screened (Cul1–5), only Cul4A/B display consistently increased neddylation ratio in $Csn2^{WT/K70E}$ pancreas (Fig. 3d). This data, together with earlier findings that Cul4 immunoprecipitates are particularly enriched for CSN subunits[36,37], suggests that partial loss of the $IP_6$ "glue" primarily releases CRL4 from CSN sequestration.

To identify downstream CRL4 modules, we next employed an in-islet viral knockdown assay to screen a collection of putative and verified CRL4 substrate receptors for insulin secretion promoter(s). This screen reveals COP1 as a prominent positive regulator of insulin secretion (Fig. 3e). Consistently, *COP1* knockdown markedly diminishes islet insulin secretion and eliminates the differences between wild-type and $Csn2^{WT/K70E}$ islets (basal or glucose-stimulated; Fig. 3f). COP1 can function as an independent E3 ligase, or as a CRL4 substrate receptor by working together with DET1 (refs. [34,35]). In the pancreas, COP1 is islet-enriched and promotes insulin secretion by degrading transcription factors ETV1/4/5 (ref. [8]), a pathway compromised in low-exocytotic β cells from T2D patients[38]. Depleting *COP1* or *DET1* leads to the accumulation of ETV5 and, to a mild extent, ETV4 in MIN6 insulinoma cells (Fig. 3g), whereas co-expressing COP1 and DET1 diminishes ETV5 levels (Fig. 3h), suggesting that CRL4$^{COP1/DET1}$ mediates ETV5 degradation, which is further supported by ETV5 stabilization upon MLN4924 treatment (Fig. 3i). Levels of ETV5, but not ETV1, ETV4, or other COP1-indpendent CRL4 substrates (CSB, Cdt1), is significant downregulated in $Csn2^{WT/K70E}$ pancreas (Fig. 3d) and in isolated $Csn2^{WT/K70E}$ islets (Fig. 3i), but can be restored to the same level as in wild-type islets upon MLN4924 treatment (Fig. 3i). The diminished ETV5 levels in $Csn2^{WT/K70E}$ mice correlates with enhanced ETV5 ubiquitylation (Fig. 3j). CSN2 knockdown similarly enhances ETV5 ubiquitylation (Fig. 3k). These data suggest that partial abolishment of CSN sequestration activates a CRL4$^{COP1}$–ETV5 degradation axis in $Csn2^{WT/K70E}$ pancreas.

ETV5 is linked to body mass index and other obesity-related traits in GWAS[10–12]. Given that obesity epidemiologically correlates with insulin levels[39–41], yet both positive and negative regulation of insulin secretion has been proposed for ETV5 (refs. [8,42]), we reinvestigated its precise role. Depleting *Etv5* enhances basal and glucose-stimulated insulin secretion in islets (Fig. 3l), whereas overexpressing *Etv5* represses basal and glucose-stimulated insulin secretion (Fig. 3m). Mechanistically, we examined two previously identified ETV transcriptional targets that are exocytosis regulators: synaptotagmin-like 3 (Sytl3) and exocyst 6 (Exoc6)[8]. Depleting *Etv5* decreases *Sytl3* and *Exoc6* transcript levels in islets, while increasing that of *Ins1* (Fig. 3n). Conversely, overexpressing *Etv5* upregulates the mRNA levels of *Sytl3* and *Exoc6*, but not *Ins1* (Fig. 3o). Both Sytl3 and Exoc6 exert anti-exocytotic effects, since their overexpression significantly inhibits insulin secretion from INS1 cells, with their combined expression having the most pronounced inhibitory effects (Fig. 3p). Further chromatin immunoprecipitation (ChIP) experiments validated ETV5 binding to Sytl3 and Exoc6 promoter regions (Fig. S2d). These data, together with diminished mRNA (Fig. 2g) and protein (Fig. 3d) levels of Sytl3 and Exoc6 in $Csn2^{WT/K70E}$ pancreas, supports an inhibitory ETV5-Sytl3/Exoc6 transcriptional axis in controlling insulin secretion[8], which is compromised in $Csn2^{WT/K70E}$ mice due to CRL4$^{COP1}$-mediated ETV5 degradation.

## COP1 outcompetes CSN to assemble the CRL4$^{COP1}$ E3 ligase in $Csn2^{WT/K70E}$ mice. The above data suggest that CSN is a haploinsufficient regulator of CRL4$^{COP1}$. However, it remains unclear why CRL4$^{COP1}$ seems particularly sensitive to CSN regulation. CSN can bind CRL4 loaded with classical WD40 domain-containing substrate receptors known as DCAFs (e.g., DDB2)[19,43], but ubiquitylation substrates sterically compete with CSN[19]. COP1 is an unorthodox substrate receptor whose binding to CRL4 is indirectly mediated by the DCAF DET1 (refs. [44,45]). Consistently, DET1 knockdown abolishes COP1 interaction with Cul4 (Fig. S3a). Using truncation and deletion variant mapping, we identified the N-terminal (aa 1–110) and C-terminal (aa 287–450) regions of DET1 mediates binding to the CRL4 adaptor DDB1 (Fig. S3c) and COP1 (Fig. S3d, e), respectively. Furthermore, exon 7 (aa 277–296) of COP1 mediates its binding to DET1 and indirectly DDB1 (Fig. S3b), but not the ubiquitylation substrate ETV5, consistent with a prior report[45]; When these information were integrated to build a structural model of CRL4$^{COP1}$, potential steric overlap between COP1 and CSN in the cryo-EM structure of CRL4–CSN[19,43] was noted (Fig. 4a). We therefore hypothesized that, similar to ubiquitylation substrates, COP1 competes with CSN for mutually exclusive CRL4 binding. Consistent with this notion, CSN can pull-down DET1, but not COP1 (Fig. 4b). Rather, co-expressing COP1 with DET1 significantly diminishes DET1 and Cul4 co-immunoprecipitation with CSN, suggesting that CRL4$^{COP1}$ and CRL4–CSN are mutually exclusive complexes because of steric incompatibility between COP1 and CSN.

We then reasoned that, in heterozygous $Csn2^{WT/K70E}$ mice with weakened CRL4–CSN interaction, the COP1–CSN competition is likely altered to favor CRL4$^{COP1}$ complex assembly. Consistent with this notion, COP1 co-immunoprecipitation with Cul4A and Cul4B is greatly increased in $Csn2^{WT/K70E}$ pancreas (Fig. 4c, d). Importantly, addition of the metabolite $IP_6$ into lysates dramatically diminishes COP1 immunoprecipitation with Cul4 (Fig. 4e), suggesting that $IP_6$ delimits CRL4–COP1 complex formation. Importantly, we found that nutrient availability reciprocally regulates the formation of CLR4$^{COP1}$ vs CRL4–CSN complexes. Thus, glucose withdrawal increases CRL4–CSN complex formation and decreases CRL4$^{COP1}$ assembly, both effects reversed by refeeding glucose containing medium (Fig. 4f, g). Together, these results suggest that direct CSN competition with COP1 for CRL4

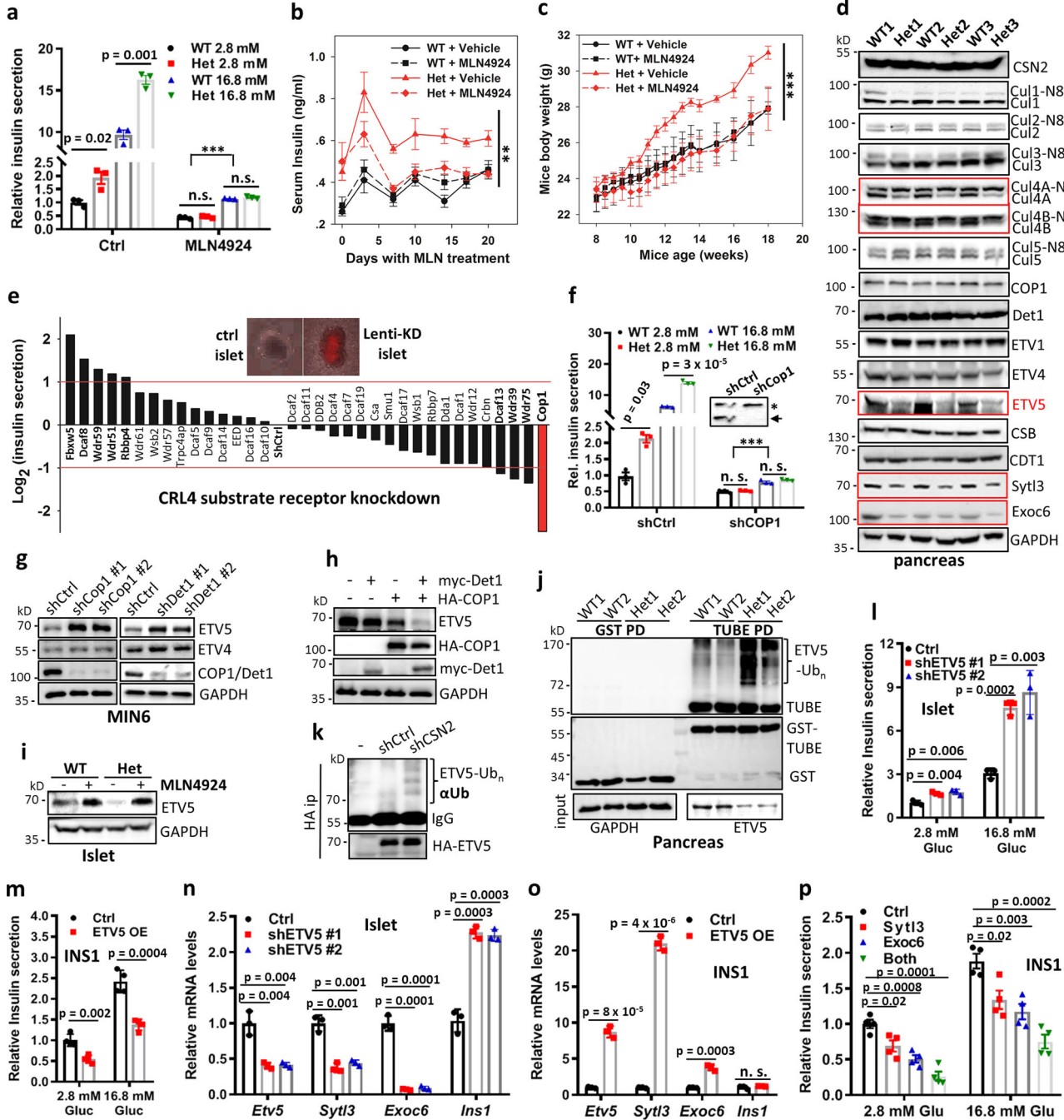

requires the $IP_6$ cofactor, whose partial absence in $Csn2^{WT/K70E}$ mice leads to CRL4$^{COP1}$ assembly.

**The CSN–CRL4$^{COP1}$–ETV5 axis is a glucose-regulated checkpoint for nutrient-induced insulin secretion.** Given the pleiotropic control of insulin homeostasis by ETV5, we investigated its physiological regulation during nutrient-stimulated insulin secretion. Glucose addition to medium promotes CRL4$^{COP1}$ complex assembly in β cells (Fig. 4f), whereas refeeding fasted mice promotes CRL4$^{COP1}$ complex assembly in pancreas (Fig. 5a), suggesting nutrient regulation of a CRL4$^{COP1}$-mediated degradation pathway. Interestingly, pancreatic levels of ETV5, but not ETV4, time-dependently accumulates upon fasting (Fig. 5b), and diminishes upon refeeding (Fig. S4a), with near-complete ETV5 degradation after 1-h refeeding in whole pancreas and in islets, evidenced by western blotting (Fig. 5c) and immunostaining (Fig. 5d), respectively. This feeding-

induced ETV5 degradation is not observed in other metabolic organs, such as liver or the white adipose tissue (Fig. S4b, c). Similar to fasting in mice, glucose withdrawal stabilizes islet ETV5 (Fig. S4d). In contrast, direct glucose application depletes ETV5 in mice (Fig. 5d) and in isolated pancreatic islets in a time-dependent manner (Fig. 5e and Fig. S4e), which is accompanied by decreased transcription of *Sytl3* and *Exoc6* (Fig. S4f). These data suggest that ETV5 is degraded in a glucose-dependent manner to relieve the inhibition on insulin secretion.

We then investigated whether glucose-induced ETV5 degradation is mediated by CRL4. The CRL inhibitor MLN4924 completely reverses feeding-induced ETV5 depletion, without altering fasting ETV5 levels, suggesting involvement of CRL4 (Fig. 5g). Consistently, elevating glucose concentrations increases Cul4 neddylation in islets (Fig. 5h). Such glucose-dependent changes in Cul4 neddylation could be attributed to diminished

**Fig. 3 Activation of the CRL4$^{COP1}$-ETV5 degradation axis underlies glucose-induced insulin secretion and is constitutively augmented in *Csn2$^{WT/K70E}$* (Het) mice. a** Effect of MLN4924 treatment (2 μM, 2 h) on basal and glucose-stimulated insulin secretion by wild-type and *Csn2$^{WT/K70E}$* islets ($n = 6$, ***$p < 0.001$, calculated by two-way ANOVA testing the effect of 2.8 or 16.8 mM glucose in the presence of MLN4924, the other $p$ values were calculated by two-tailed Student's $t$ test). Data are presented as mean ± SEM. **b**, **c** Mice serum insulin levels with/without MLN4924 treatment (15 mg/kg, twice weekly for 6 weeks). Insulin levels were measured at the same time of the day to minimize fluctuation. ($n = 6$, **$p = 0.001$, calculated by two-way repeated measures ANOVA testing genotype × time effect for Het mice). Data are presented as mean ± SEM. **c** Mice body weight with/without MLN4924 treatment (15 mg/kg, twice weekly for 6 weeks; $n = 6$, ***$p < 0.001$, calculated by two-way repeated measures ANOVA testing genotype × time effect for Het mice). Data are presented as mean ± SEM. **d** Levels of total and neddylated Cullins, COP1, and CRL4 substrates in the pancreas after 6 h fasting to normalize variation in blood glucose levels. **e** Ex vivo, lentiviral shRNA-based knockdown screening for CRL4 adaptors whose depletion alters islet insulin secretion. Quantified values of insulin secretion were normalize by that of Ctrl islets and presented in Log$_2$ scale. **f** Effect of adenovirus-mediated in-islet COP1 knockdown on basal and glucose-stimulated insulin secretion ($n = 3$, ***$p < 0.001$, calculated by two-way ANOVA testing the effect of 2.8/16.8 mM glucose in the presence of shCOP1, the other $p$ values were calculated by two-tailed Student's $t$ test). Data are presented as mean ± SEM. Inset: western blot showing efficient COP1 knockdown. Arrow indicates COP1 band. An asterisk (*) indicates nonspecific band. **g** Levels of ETV4 and ETV5 in MIN6 cells with COP1 (left) or Det1 (right) knockdown using two independent shRNA. **h** Levels of ETV5 in MIN6 cells with Myc-Det1 and HA-COP1 single or double overexpression. **i** Levels of ETV5 in wild-type and *Csn2$^{WT/K70E}$* islets with/without MLN4924 treatment (2 μM, 2 h). **j** Levels of ubiquitylated ETV5 enriched by GST-TUBE pulldown. **k** Effect of CSN2 knockdown on ETV5 ubiquitylation in MIN6 cells. **l** Effect of adenovirus-mediated in-islet ETV5 knockdown on insulin secretion ($n = 3$). Data are presented as mean ± SD. $P$ values were calculated by two-tailed Student's $t$ test. **m** Effect of ETV5 overexpression on insulin secretion ($n = 4$). Data are presented as mean ± SD. $P$ values were calculated by two-tailed Student's $t$ test. **n** Effect of adenovirus-mediated in-islet ETV5 knockdown on mRNA levels of ETV5 transcriptional targets ($n = 3$). Data are presented as mean ± SD. $P$ values were calculated by two-tailed Student's $t$ test. **o** Effect of ETV5 overexpression on mRNA levels of ETV5 transcriptional targets ($n = 3$). Data are presented as mean ± SD. $P$ values were calculated by two-tailed Student's $t$ test. **p** Effect of Sytl3 and Exoc6 overexpression on insulin secretion ($n = 3$). Data are presented as mean ± SEM. $P$ values were calculated by two-tailed Student's $t$ test.

CSN-mediated deneddylation, since CRL–CSN interaction is stronger under fasting and is diminished upon glucose supplementation (Fig. 4f), although total levels of IP$_6$ remain unchanged between fasted and fed mice (Fig. S4g). Together, these data together suggest that glucose-induced assembly of CRL4$^{COP1}$ destabilizes ETV5.

In summary, the data presented so far suggest that CRL4$^{COP1}$ liberation from IP$_6$-dependent CSN sequestration and consequent ETV5 degradation relieves the checkpoint for nutrient (glucose)-induced insulin secretion, a process constitutively augmented in *Csn2$^{WT/K70E}$* mice.

**The CRL4$^{COP1}$-ETV5 axis in human islets and β cells.** Given the GWAS-identified ETV5 association with diabetes traits[10–12], and the compromised COP1 activity in β cells from T2D patients[38], we also examined the relevance of the CRL4$^{COP1}$-ETV5 pathway in human using isolated islets and the EndoC-βH1 human β cells[46]. CRL inhibition and COP1 knockdown both decrease insulin secretion from isolated islets and EndoC-βH1 cells, more prominently under high glucose conditions, when CRL4$^{COP1}$ is assembled and active (Fig. 6a, b, e, f). By contrast, ETV5 knockdown enhances insulin secretion (Fig. 6c, g). Consistently, GSIS is accompanied by the depletion of ETV5 levels from isolated islets and EndoC-βH1 cells (Fig. 6d, h). These data suggest that the CRL4$^{COP1}$-ETV5 axis also operates during insulin secretion in human.

**CRL neddylation inhibition prevents ETV5 degradation and hyperinsulinemia in diet- and leptin deficiency-induced obesity/diabetes.** One emerging approach to treat obesity and insulin resistance is mild suppression of hyperinsulinemia[47]. The CRL inhibitor MLN4924 specifically inhibits hyperinsulinemia in *Csn2$^{WT/K70E}$* mice, without significantly affecting serum insulin levels in wild-type littermates (Fig. 3b). Given that the pancreatic CRL4$^{COP1}$-ETV5 axis is hyperactive not only in *Csn2$^{WT/K70E}$* mice (Fig. 3), but also in HFD-fed mice (Fig. 7a), we examined whether CRL inhibition could rescue disease progression in this obesity/diabetes model. Indeed, MLN4924 attenuates HFD-induced ETV5 degradation (Fig. 7a), obesity (Fig. 7b), hyperinsulinemia (Fig. 7c), hyperglycemia (Fig. 7d), glucose intolerance (Fig. S5b), and insulin resistance (Fig. S5c). In comparison, in

mice fed with normal chow diet (NCD), levels of ETV5 (Fig. S5a), blood glucose (Fig. 7c), serum insulin (Fig. 7d), glucose clearance (Fig. S5b), and insulin sensitivity (Fig. S5c) are not significantly affected by the same regime of MLN4924 treatment. While the effect of MLN4924 maybe global[48], mimicking hyperinsulinemia via chronic administration of long-acting insulin (Glargine®) reverses the inhibitory effect of MLN4924 on HFD-induced weight gain (Fig. S5d), suggesting that MLN4924 inhibits diet-induced obesity at least in part via inhibiting insulin hypersecretion.

To exclude possible HFD-specific effects of MLN4924, we examined the leptin-deficient ob/ob mice, whose uncontrolled food intake elicits hyperinsulinemia and low ETV5 levels under normal diet (Fig. 7e). MLN4924 also attenuates excessive ETV5 degradation (Fig. 7e), obesity, and diabetes development in ob/ob mice (Fig. 7f), improving their hyperglycemia (Fig. 7g), hyperinsulinemia (Fig. 7h), glucose tolerance, and insulin sensitivity (Fig. S5e, f). Together, these data suggest that CRL inhibition is a plausible treatment option for hyperinsulinemia, diet/overeating-induced obesity, and obesity-associated diabetes (Fig. 8).

**Discussion**

Compared to mechanisms facilitating glucose-induced insulin release, less is known about β-cell autonomous regulators that restrict unwanted or untimely insulin production and secretion. In dissecting the physiological implications of metabolite (IP$_6$)-mediated CSN–CRL interaction, we demonstrate a hitherto unknown role for mammalian CSN in metabolic control. CSN regulates pancreatic insulin secretion via sequestrating CRL4 from CRL4$^{COP1}$ E3 ligase assembly, leading to the build-up ETV5, a transcriptional "brake" that restricts insulin secretion and is relieved upon glycemic rise (Fig. 8). CRL4$^{COP1}$ also plays other roles in development and differentiation[9,49], whether glucose regulates CRL4$^{COP1}$ in these contexts remain to be investigated. In plants, COP1 and CSN both mediate light-controlled growth via interacting with the CRL4 E3 ligase[50,51]. While it would be interesting to examine whether COP1–CSN competition also determines the assembly and E3 ligase activity of CRL4$^{COP1}$ in plants, fundamental interaction between COP1 and CSN in growth and metabolism seems preserved across evolution.

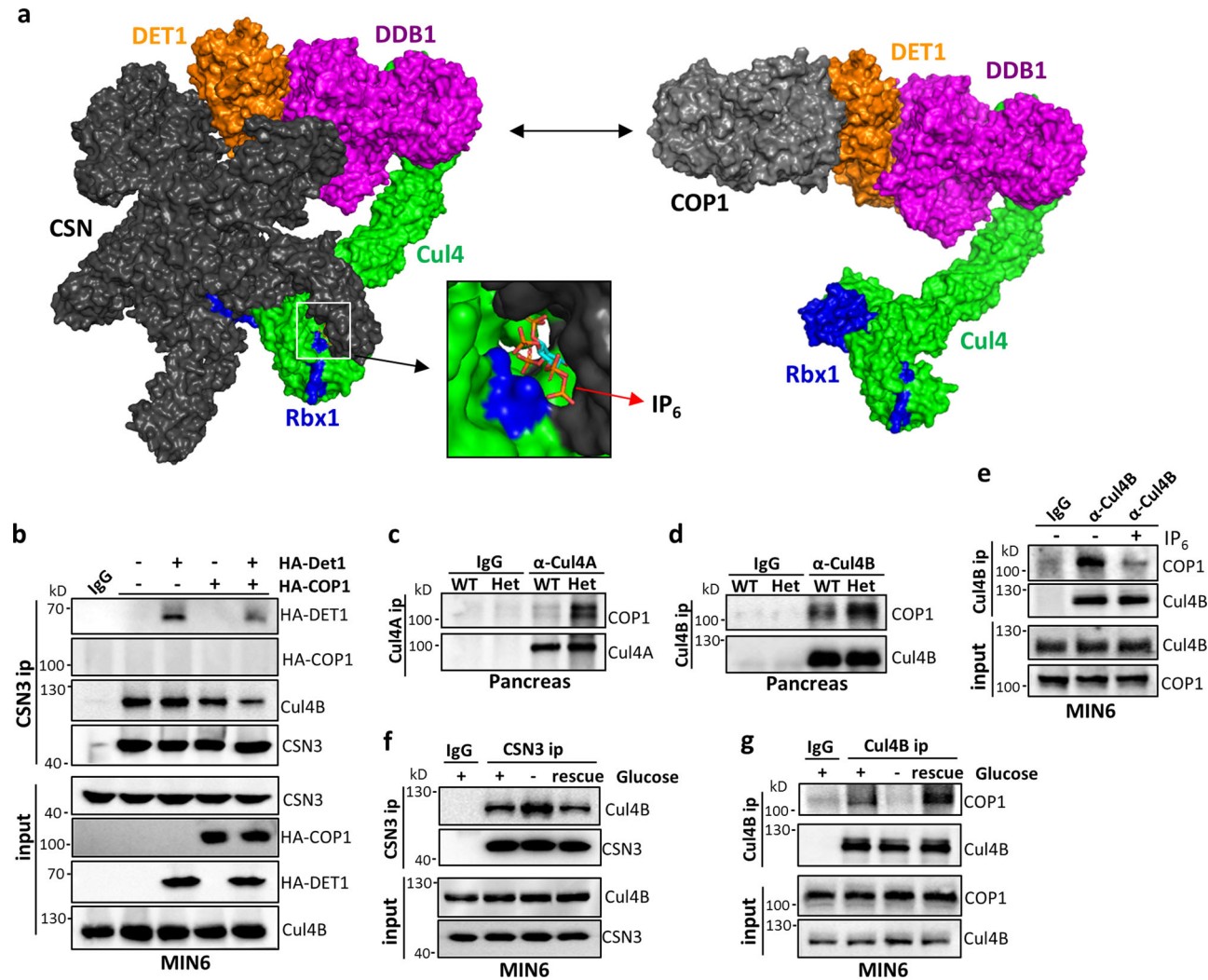

**Fig. 4 COP1 outcompetes CSN to assemble the CRL4$^{COP1}$ E3 ligase for ETV5 degradation in $Csn2^{WT/K70E}$ (Het) mice and in response to nutrients. a** Structural models illustrating steric incompatibility between COP1 and CSN. For modeling details, see "Methods". **b** Effect of COP1 and DET1 overexpression, individually or combined, on CRL–CSN complex formation in MIN6 cells. **c**, **d** CRL4–COP1 complex formation detected by COP1 co-immunoprecipitation with Cul4A (**c**) and Cul4B (**d**) from WT and $Csn2^{WT/K70E}$ pancreas lysates. **e** Effect of adding IP$_6$ (20 μM) into MIN6 cell lysates on CRL4B$^{COP1}$ complex formation. **f** Glucose fasting (2 h) increases CSN3-Cul4B co-ip in MIN6 cells, which is reversed by replacement with glucose-proficient medium (1 h). **g** Glucose fasting (2 h) decreases Cul4B-CSN3 co-ip in MIN6 cells, which is reversed by replacement with glucose-proficient medium (1 h).

The CSN cofactor IP$_6$ directly interacts with the Rbx1 subunit of CRL, not Cullins, thus potentially influencing all Cullin members that heterodimerizes with Rbx1/2 (refs. [25]). Such broad impact of IP$_6$ could possibly account for embryonic lethality in $Csn2^{K70E/K70E}$ homozygous mice[25]. In the $Csn2^{WT/K70E}$ heterozygous mice, however, partial abolishment of the IP$_6$ cofactor has rather specific influences on the pancreatic CRL4$^{COP1}$–ETV5 axis at multiple levels. First, only Cul4A/B display significantly enhanced neddylation, which is likely due to larger percentage of Cul4s being CSN-bound during their catalytic cycle[29,30], rendering them sensitive to even partial CSN dysfunction. Given that Cul4A and Cul4B are both hyper-neddylated, and that assembly of CRL4A$^{COP1}$ and CRL4B$^{COP1}$ E3 ligases are both increased in $Csn2^{WT/K70E}$ mice, it remains to be determined if CRL4A$^{COP1}$ and CRL4B$^{COP1}$ are equally involved in ETV5 ubiquitylation. Second, among the CRL4s, the assembly of CRL4$^{COP1}$ is prominently upregulated in $Csn2^{WT/K70E}$ pancreas. This is because COP1 is islet-enriched[8], where it partners with DET1 to compete with CSN for free CRL4, unlike canonical substrate receptors (e.g., DDB2, CSA) that coexist with CSN-bound CRL4 (ref. [19]).

Finally, ETV5, but not other CRL4$^{COP1}$ substrates examined, is degraded in the $Csn2^{WT/K70E}$ mutant mice and, more physiologically, in fasted wild-type mice upon refeeding. The same treatment does not deplete ETV5 levels in liver or adipose tissue, pointing toward a highly dedicated IP$_6$–CSN–CRL4$^{COP1}$–ETV5 pathway in pancreatic nutrient response, whereby glucose-induced CRL4$^{COP1}$ assembly and yet-unknown modifications on ETV5 may serve as coincidental triggers for its recognition and degradation. Given that ETV5 degradation is nutrient-regulated in β cells and in whole pancreas, its role in other cell types worth further investigation.

The clinical relevance of ETV5 as a CRL4$^{COP1}$-degradable checkpoint in feeding-induced insulin secretion is underscored by GWAS identification of ETV5 as a diabetes/obesity-related gene[10–12]. GWAS-identified gene associated with diabetes are increasingly appreciated to relate to islet function[13]. Given the pronounced assembly of CRL4$^{COP1}$ E3 ligase upon feeding, other COP1 targets may also be ubiquitylated in a stimuli-dependent context, thereby contributing to glucose homeostasis regulation. In this regard, the COP1 degradation axis has been recently

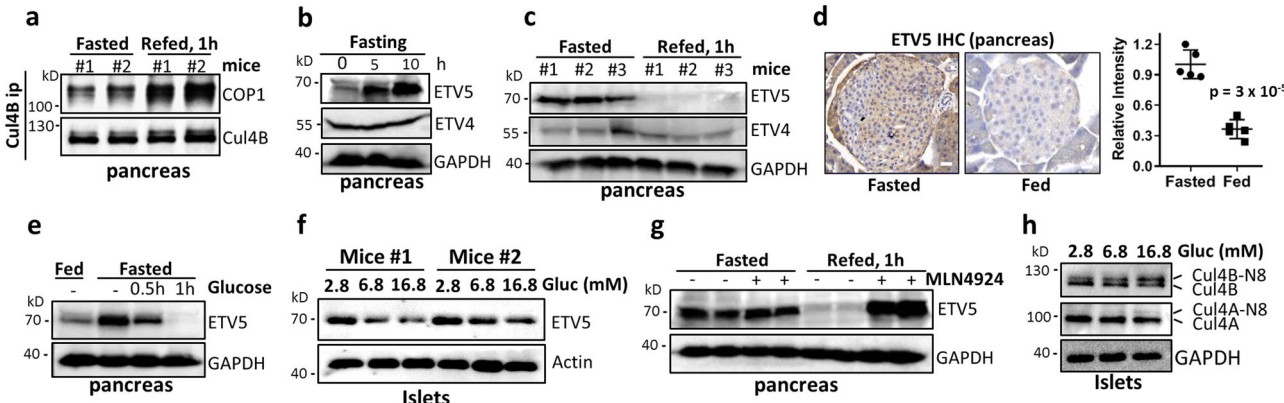

**Fig. 5 ETV5 as a CRL4^COP1 regulated physiologic checkpoint for nutrient/glucose-induced insulin secretion. a** Refeeding induces CRL4B–COP1 complex formation in overnight-fasted mice. **b** Levels of ETV4 and ETV5 in the pancreas of mice fasted for the indicated period. **c** Levels of ETV5 in the pancreas of overnight-fasted mice, with/without refeeding. **d** Immunostaining of ETV5 in pancreatic tissues of fasted or fed mice ($n = 5$, ***$p < 0.001$, Student's $t$ test). Data are presented as mean ± SEM. **e** Effect of glucose (1.5 mg/kg) on ETV5 levels in pancreas. Scare bars, 0.2 μm. **f** Levels of ETV5 in isolated islets after exposure to different concentrations of glucose for 1 h. **g** Levels of ETV5 in the pancreas of starved/refed mice with/without MLN4924 pretreatment (15 mg/kg, 1 h). **h** Increasing glucose concentration in culture medium promotes Cul4 neddylation in islets.

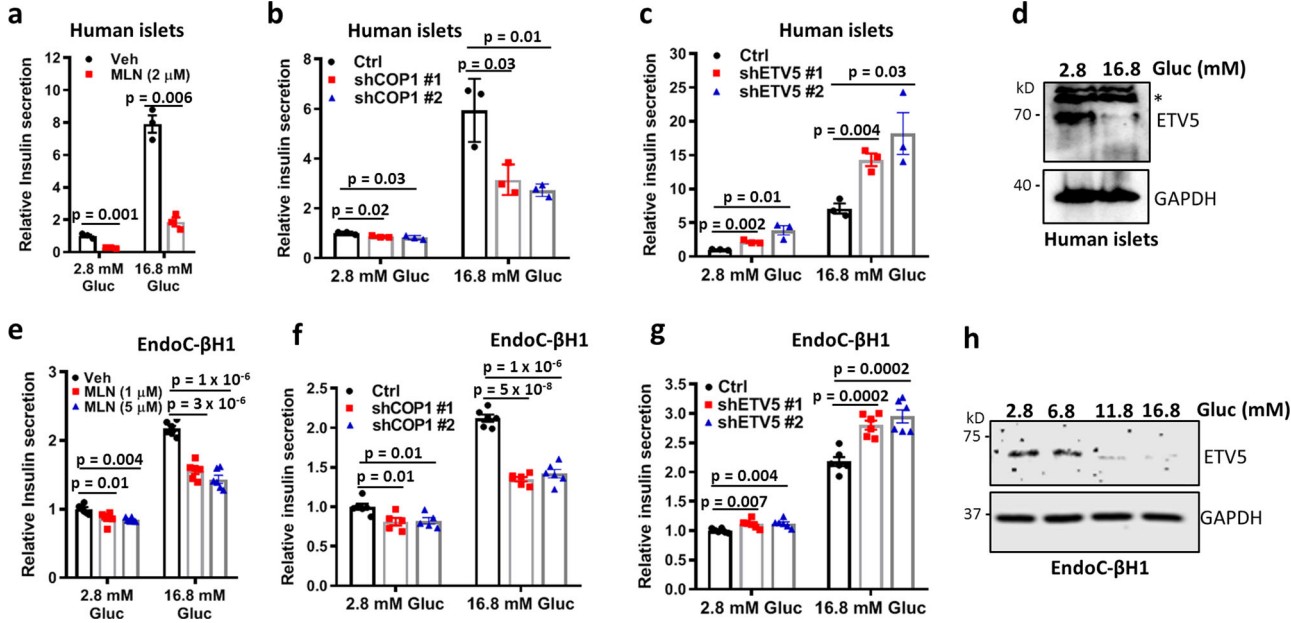

**Fig. 6 The CRL4^COP1–ETV5 axis operates in human islets and β cells. a–c** Effects of CRL inhibition (**a**), and COP1 (**b**), or ETV5 (**c**) knockdown on insulin secretion from isolated human islets ($n = 3$). Data are presented as mean ± SEM. $P$ values were calculated by two-tailed Student's $t$ test. **d** High glucose medium depletes ETV5 levels in isolated human islets. An asterisk (*) indicates a nonspecific band. **e–g** Effects of CRL inhibition (**e**), and COP1 (**f**) or ETV5 (**g**) knockdown on insulin secretion from the EndoC-βH1 human beta cells ($n = 6$). Data are presented as mean ± SEM. $P$ values were calculated by two-tailed Student's $t$ test. **h** High glucose medium depletes ETV5 levels in EndoC-βH1 cells.

shown to malfunction in T2D patient β cells with low-exocytotic capacity, whereby upregulated ETV1 suppresses insulin secretion via an inflammatory pathway[38], suggesting a COP1-ETV1 axis preventing β cell failure at later stages of T2D. These insights provide a new window to explore the etiology of potential disease-associated variants of COP1 and ETVs.

Two tightly linked and co-evolving hallmarks of diet-induced obesity and diabetes are insulin resistance and hyperinsulinemia[52], the cause and consequence between them remain a subject of discussion[5,30,47]. Mice with reduced insulin gene dosage are resistant to diet-induced obesity and insulin resistance[53], while mice carrying extra copies of insulin gene develop insulin resistance as a consequence of hyperinsulinemia[54]. The $Csn2^{WT/K70E}$ mice we generated

represent a rare nontransgenic model of "congenital hyper-insulinism", whereby insulin oversecretion drives resistance. These observations, together with the notable correlation between insulin levels and obesity/diabetes in humans[39–41], indicate that mild suppression of hyperinsulinemia can be therapeutic. Indeed, lowering insulin levels with β-cell $K_{ATP}$ channel openers has proved beneficial in rodents and humans[55]. However, the challenge remains to identify agents that selectively inhibits hyperinsulinemia without affecting normal insulin secretion[47]. Given that the general CRL inhibitor MLN4924 efficiently inhibits hyperinsulinemia, but not normal insulin levels or sensitivity, and that CRL4^COP1 is particularly sensitive to partial disruption of CSN regulation, more specific modulation of insulin secretion via targeting

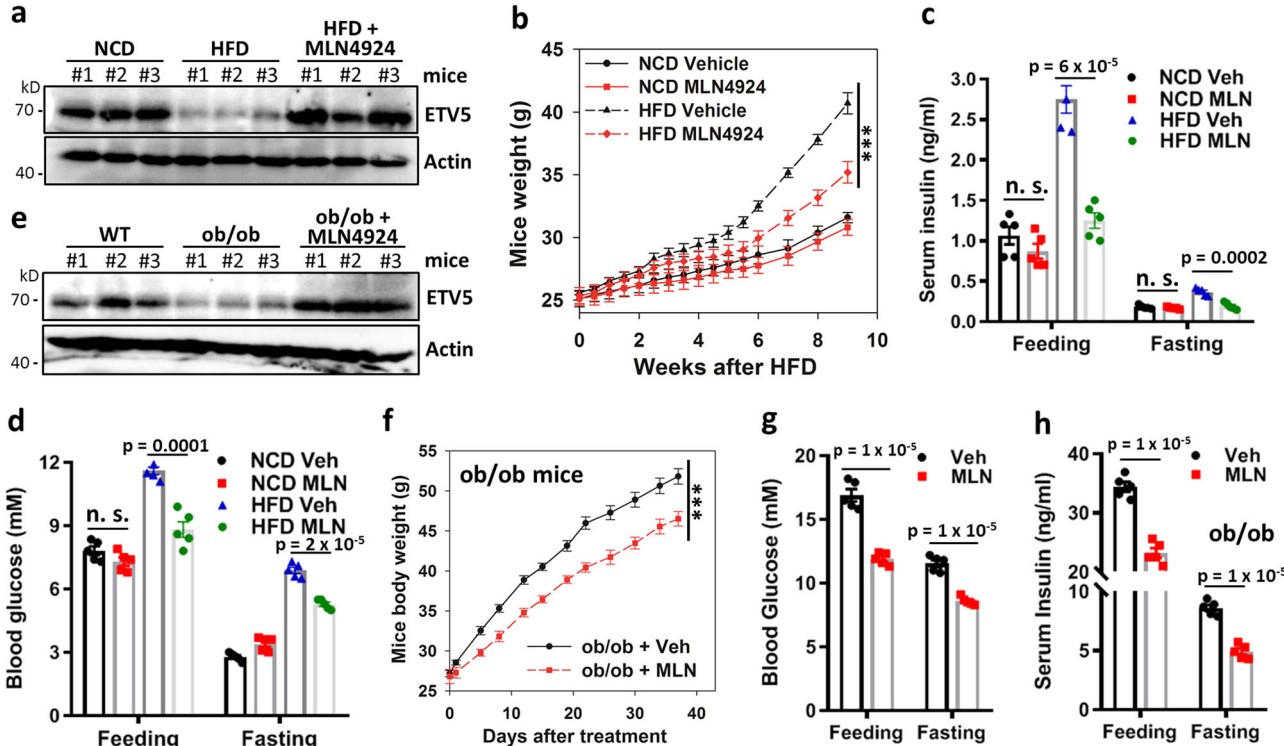

**Fig. 7 The CRL inhibitor pevonedistat/MLN4924 stabilizes ETV5 and improves diet-induced or leptin deficiency-induced obesity/diabetes. a**, **e** Levels of ETV5 in HFD-fed mice (**a**) or ob/ob mice (**e**), with/without MLN4924 treatment. **b** Body weight of 8-week-old WT mice fed HFD or NCD for the indicated time period, with/without MLN4924 treatment (15 mg/kg, twice weekly for the first 6 weeks; $n = 5$, ***$p < 0.001$, calculated by two-way repeated measures ANOVA testing treatment × time effect). Data are presented as mean ± SEM. **c**, **d** Feeding and fasting insulin (**c**) and glucose (**d**) levels in HFD or NCD mice with/without MLN4924 treatment ($n = 5$). Data are presented as mean ± SEM. $P$ values were calculated by two-tailed Student's $t$ test. **f** Body weight of ob/ob mice with/without MLN4924 treatment (15 mg/kg, twice weekly; $n = 5$, ***$p < 0.001$, calculated by two-way repeated measures ANOVA testing treatment × time effect). Data are presented as mean ± SEM. **g**, **h** Feeding and fasting glucose (**g**) and insulin (**h**) levels in ob/ob mice with/without MLN4924 treatment ($n = 5$). Data are presented as mean ± SEM. $P$ values were calculated by two-tailed Student's $t$ test.

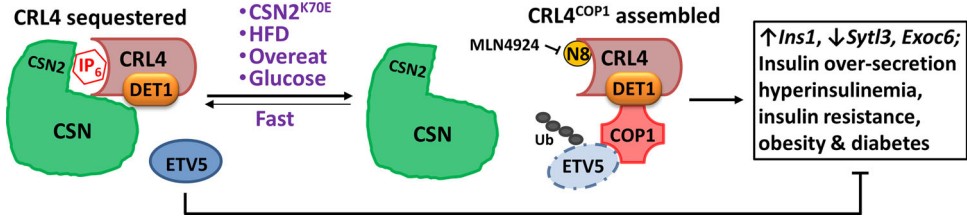

**Fig. 8 Scheme depicting the IP$_6$–CSN–CRL4$^{COP1}$–ETV5 pathway as a checkpoint controlling insulin secretion and its potential role in the development of diet/overeating-induced obesity and diabetes.** CRL4-sequestration by CSN is compromised by CSN2-K70E mutation, or hyperglycemia elicited by nutrient oversupply. This leads to the assembly of CRL4$^{COP1}$ E3 ligase to promote ETV5 degradation, which acts as a transcriptional "brake" suppressing congenital hyperinsulinemia and consequent obesity/diabetes.

Cul4 regulators that determine CRL4$^{COP1}$ assembly, including the inositol polyphosphate pathway, may benefit obese and early T2D patients.

## Methods

**Materials**. HEK 293, 293T, MIN6, and INS1 cells were purchased from ATCC and cultured in DMEM/10% FBS/1% glutamine/1% Penstrep at 5% $CO_2$ (vol/vol). The primary antibodies used were: Cul1 (1:1000, CST, 4995), Cul3 (1:1000, CST, 2759), CSN5 (1:1000, CST, 6895), CDT1 (1:1000, Proteintech, 14382-1-AP); ETV1 (1:1000, Abcam, ab184120), ETV5 (1:1000, Abcam, ab102010, CHIP and WB), Cul4A (1:1000, Abcam, ab72548, WB and IP), CSN3 (1:1000, Abcam, ab79698), Cul2 (1:1000, Bethyl, A302–476A), Cul5 (1:1000, Bethyl, A302–173A), COP1 (1:1000, Bethyl, A302–173A), CSB (1:1000, Genetex, GTX104589), GAPDH (1:3000, Proteintech, 10494-1-AP), Exoc6 (1:1000, Proteintech, 12723-1-AP), Sytl3 (1:1000, Proteintech, 22076-1-AP), CSN2 (1:3000, Proteintech, 10969-2-AP), ETV4 (1:1000, Thermo, MA5-15424), Cul4B (1:1000, Sigma, C9995, WB and IP), GST (1:3000, Sigma, A7340), Det1 (1:1000, Santa Cruz, sc-514348), insulin (1:100, R&D Systems, I2018, for IF), and glucagon (1:100, Abcam, ab10988, for IF). Trypsin, shRNA for

CSN2 (id: 1773s21c1), and scrambled shRNA were purchased from Sigma. The other shRNAs were generated by cloning into pLKO.1 vector. The corresponding primers are synthesized by Sangon Biotech and are listed in the Supplementary Table 1, which also contain other primers used in this study. IgG beads were purchased from GE Healthcare. GST-TUBE was from LifeSensors. Premade insulin ELISA kit was from Crystal Chem. Glucose-free DMEM and RPMI are from Gibco. All other reagents were purchased from Sigma unless otherwise specified.

**GST pulldown and co-immunoprecipitation**. Cells were lysed in lysis buffer (50 mM Tris-HCl, pH 7.5, 125 mM NaCl, l% Triton X-100, 1× phosphatase inhibitor, and 1× protease inhibitor). For GST pulldown experiments, cell lysate was incubated with glutathione beads at 4 °C for 2 h. For immunoprecipitation experiments, cell lysate was first incubated with desired antibody at 4 °C overnight, followed by incubation with protein A/G plus beads (Santa Cruz, Inc) at 4 °C for another 1 h. Beads were then washed three times with lysis buffer and boiled in SDS sample buffer. Samples were loaded to SDS–PAGE gel for western blotting of the indicated proteins. Images were acquired using ChampChemi 610 Chemiluminescence Analysis System (SageCreation).

**IP$_6$ detection**. The purification and visualization of IP$_6$ from pancreas were performed according to previously established methods[25,56,57]. Tissues were hmogenized and extracted in 1 M perchloric acid (PA) containing 5 mM EDTA. After votexing for 10 min, samples were centrifuged at $18,000 \times g$ for 5 min at 4 °C. The titanium dioxide (TiO$_2$) beads (Titansphere TiO 5 μm;GL Sciences) were prepared by washing in water and then in 1 M PA containing 5 mM EDTA (4–5 mg for one sample). The supernatants from tissue lysate were mixed with TiO$_2$ beads added by rotation for 15 min at 4 °C. After centrifuging at $3500 \times g$ for 1 min, beads were washed twice in PA. Bound inositol phosphates were then eluted with 200 μl 5% ammonium hydroxide, which were then vacuum evaporated to 50 μl and subjected to 35.5% PAGE. Phosphate-rich metabolites were imaged with toluidine blue staining, using commercial IP$_6$ as a control.

**Chromatin immunoprecipitation**. ETV5 ChIP was performed as previously described[58]. Briefly, INS1 cells were crosslinked with formaldehyde, and lysed in lysis buffer [10 mM Tris, 10 mM NaCl, 0.2% NP40 (Ph8.0), and protease inhibitors]. Nucleus were extracted in 1% SDS and sheared to an average length of 300–500 bp by sonication (Qsonica Q700, 6 pulses, 15 s on, 45 s off at 5 W), followed by dilution in i.p. buffer and overnight incubation with ETV5 antibody (ab102010, 1:100). Upon protein A/G-agarose beads precipitation and wash, immunoprecipitated DNA were de-crosslinked by heating (95 °C, 15 min). Q-PCR was performed using SYBR ® Green Real-time PCR Master Mix, with the following primers: Sytl3-F:5′-ACTCAAACTGGGCTCCCAAG-3′, Sytl3-R:5′-AATGAGAAT CTGTACCTAGA-3′; Exoc6-F:5′-AATATCATCACCCAGGATTC-3′, Exoc6-R:5′-GTCAGGCAGGCTTTGCACTA -3′.

**Mouse genetics**. All mice were maintained in specific pathogen-free facilities in the Laboratory Animal Research Center of Southern University of Science and Technology. Mice were housed in colony cages at 25 °C with 12-h light/12-h dark cycles, with 20–60% humidity. The dark cycle began at 7 p.m. Experiments were conducted following the SUSTech Guide for the Care and Use of Laboratory Animals, and approved by the Animal Ethics Committee (Approval No: SUSTC-2017-076). The knockin mouse model was generated in a C57BL/6J background by targeting the mouse Csn2 gene and introducing the mutation into the equivalent position (K70E) in exon 3 by CRISPR/Cas9. Genotyping was performed using specific primers against mutant allele and WT allele, respectively, followed by restriction digestion and/or DNA sequencing. Heterozygous mutant mice were backcrossed to C57BL/6J wild-type mice. Experiments were conducted exclusively on littermates and, unless specified otherwise, male mice were used for analysis, although female Het mice also display heavier weight and hyperinsulinemia. Ob/ob mice were purchased from Model Animal Research Center, Nanjing University.

**HFD treatment**. At the age of 8 weeks, mice were fed with HFD (D12492, Research Diets), which consists of Rodent Diet with 60% kcal fat. HFD treatment lasted for 9 weeks, with mice's body weights, blood glucose levels monitored weekly. MLN4924 (ApexBio) was dissolved in (2-hydroxypropyl)-β-cyclodextrin (Vehicle) and injected intraperitoneally.

**Measurement of glucose, FFA, TG, and insulin levels**. Blood samples are taken by tail vein nick, the glucose levels are measured by glucometer (Contour TS, Bayer), and levels of insulin, nonesterified free fatty acids, and triglycerides are detected with serum samples, using commercially available kit.

**Glucose tolerance test**. Mice were fasted for 16 h, and injected intraperitoneally with glucose (1.5 g/kg body weight). Blood was longitudinally collected at 0, 5, 10, 15, 30, 45, 60, 90, and 120 min from the same mice to measure glucose levels with a glucometer.

**Insulin tolerance test**. Mice were starved for 4–6 h prior to the ITT, Recombinant Human Insulin (ProSpec) was intraperitoneally injected (0.75 U/kg body weight). Blood was then longitudinally collected at 0, 15, 30, 45, 60, 90, and 120 min from the same mice to measure glucose levels.

**Streptozocin treatment**. Ten-week-old mice were intraperitoneally injected with 50 mg/kg Streptozocin [Sigma, 10 mg/ml in 0.1 M Sodium Citrate Buffer (pH 5.5)] for four consecutive days. Mice's body weight and blood glucose levels were then monitored weekly to ensure successful destruction of β cells, ITT was performed 6 weeks after streptozocin treatment.

**Glucose-stimulated insulin secretion assay**. Mice were fasted for at least 16 h prior to the GSIS assay, followed by glucose (1.5 g/kg body weight) injection described for the GTT. Blood samples (>20 μl) are longitudinally collected from the same mice at 5, 10, 20, and 30 min after injection. Blood was allowed to clot at room temperature for 15–30 min, spun at $2000 \times g$ in a tabletop centrifuge for 20 min at 4 °C, and the supernatant serum was collected for Insulin measurement using Insulin ELISA Kit (Crystal Chem).

**Free cytoplasmic Ca$^{2+}$ recording**. Ca$^{2+}$ recording and analysis followed published methods[8]. Briefly, islets cultured overnight on ploy-L-lysine-coated glass dishes were transferred to in REC buffer (119 mM NaCl, 2.5 mM CaCl$_2$, 4.7 mM KCl, 10 mM Hepes,1.2 mM MgSO$_4$, 1.2 mM KH2PO$_4$, and 2.8 mM glucose) with 5 μM FURA2 (invitrogen) and for 30 min. They were then put into fresh REC buffer to record basal calcium. At 3 min, 20 mM glucose or 25 mM KCl was injected to stimulate calcium influx. The A340 nm/A380 nm ratio was recorded every 2 s using the Metafluor system (Molecular Device).

**Isolated islets insulin secretion assay**. The protocol described by Suriben et al. was adapted for measuring insulin secretion from isolated islets[8]. Briefly, pancreas was frist distended and then digested in Collagenase XI/HBSS solution at 37.54 °C for 15 min, with shaking every 2–3 min. Upon digestion termination by CaCl2/HBSS solution, cells were pelleted, resuspended, and the islets were captured by filtration through a 70 μm cell strainer. After overnight recovery in RPMI medium containing 11.8 mM glucose, 10% FBS, 50 U/ml penicillin, and 50 μg/ml streptomycin, ten islets per mouse were transferred into Kreb's buffer containing 2.8 mM glucose for 1 h to measure basal insulin secretion. After supernatant removal, secretogogues (16.8 mM glucose or 20 mM arginine in 2.8 mM glucose) in Kreb's buffer were added to the islets for 1 h. Insulin in the supernatants was measured as described above. Results are presented as total insulin secreted after normalization to basal secretion in 2.8 mM glucose.

For time-split GSIS measurement, insulin concentrations detected from each time point were normalized to a concentration with total volume being 1 ml, and prior insulin levels were subtracted.

For shRNA-based screening of CRL4 adaptors regulating insulin secretion, the primer sequences are described in Supplementary Table 1. Validated shRNAs in pLKO.1 vectors were used to package lentiviral particles in HEK293T cells cultured in 10-cm plates as previously described[25]. Lentiviruses in the medium were pelleted by centrifugation and resupended in 200 μl islet culture medium, before being added to islets isolated on the same day. After infection for 24 h, viruses were removed and islets were cultured for another 24 h. Afterward, GSIS was performed as described above.

**Islet electron microscopy**. Islets were harvested from 3-month-old mice by collagenase digestion and hand picked. Samples were fixed in the 1/2 Karnovsky's Fixative (2% paraformaldehyde and 2.5% glutaraldehyde in PBS) overnight, post fixed in 1% aqueous osmium tetroxide in PBS for 2 h at 4 °C, and dehydrated through a series of ethanol washes followed by two propylene oxide washes. Samples were embedded in Eponate 12 (TEPELLA-18010) and cured at 60 °C overnight. Ultrathin (80 nm) sections were obtained with an Ultracut microtome (Leica). The ultrathin sections were stained with 1% uranyl acetate in methanol for 5 min, and washed with ddH$_2$O three times. After drying with filter paper, samples were stained with 0.2% lead citrate for 5 min, then washed with ddH$_2$O three times, and dried in the dish overnight. Sections were examined in a transmission electron microscope (Hitachi TEM-HT7700) at 100 kV.

**Pancreas morphology analysis**. Pancreas was dissected rapidly when the mouse was sacrificed, and fixed overnight in 4% paraformaldehyde. Paraffin sections were blocked in 10% horse serum, and stained with different methods including H&E staining, Aldehyde fuchsin staining (Huayueyang Biotechnology), Immunohistochemsitry (ETV5) and immunofluorescence staining. In IF staining, islets were stained for insulin (R&D Systems) and glucagon (Abcam). And the stained area was quantified using Image-Pro Plus software (Media Cybernetics).

**Human islets experiments**. The study was approved by the Ethics Committee of The Third People's Hospital of Shenzhen. Pancreatic tissues from metabolically healthy donors were donated via a dedicated Organ Procurement Organization in a double-blinded manner. Informed consents were obtained from donor's next of kin authorized to give consent. The islet isolation procedures were similarly as for mouse pancreas. Human islets were cultured in RPMI 1640 media (5.5 mM glucose and 10% FBS) and processed for GSIS with/without MLN4924 treatment or COP1 and ETV5 knockdown, with five islets per sample. To determine the effect of glucose on islet ETV5 levels, 50 islets each were cultured in Kreb's buffer containing either 2.8 or 16.8 mM glucose for 1 h, before being assayed for ETV5 by western blotting.

**EndoC-βH1 cell experiments**. EndoC-βH1 cells (EndoCells, Paris, France) were cultured as previously described[46]. Briefly, cells were cultured in DMEM containing 5.6 mM glucose, 2% BSA fraction V (Roche), 10 mM nicotinamide (Merck Millipore), 50 μM 2-mercaptoethanol, 5.5 μg/ml transferrin, 6.7 ng/ml sodium selenite (Sigma-Aldrich), and 1% (vol/vol) penicillin and streptomycin (Thermo Fisher Scientific), and were maintained at 37 °C in air with 5% CO$_2$.

For insulin secretion assay, cells were preincubated in Krebs-Ringer bicarbonate buffer (KRBB; supplemented with 10 mmol/l HEPES and 2 mg/ml BSA) with 2.8 mmol/l glucose for 2 h. After that, cells were treated with MLN4924 (0, 1, and 5 μM) for 2 h followed by incubation in KRBB containing 2.8 mM or 16.8 M glucose for an additional 1 h. Insulin release into KRBB was measured by ELISA kits. For COP1 and ETV5 knockdown experiments, EndoC-βH1 cells were infected

with the respective lentivirus and subjected to puromycin (1 μg/ml) selection for stable knockdown cell lines, which were then used for insulin secretion assays.

**Modeling**. Structural models of DET1 (Uniprot id: Q7L5Y6) and COP1 (Uniprot id: Q8NHY2) were generated by homology modeling using I-TASSER (version 5.1)[59]. After that, a model of the CRL4[DET1/COP1] complex was manually built based on experiments that determined the binding interface between DET1 and DDB1, and between DET1 and COP1 (Fig. S3). Thus, DET1 was positioned into the CRL4–CSN complex by replacing DDB2 in the cryo-EM structure of CRL4[DDB2]–CSN[19], with the helical motif of DET1 (aa 14–26) aligning to DDB1-binding helical motif of DDB2 (ref. [60]). COP1 was positioned based on prior findings that its coiled-coil domain-containing exon 7 (aa 277–296) mediates binding to DET1 (ref. [45]). Images were preparaed in Pymol (version 2.2.0), as previously described[25].

**Statistics and reproducibility**. All results are presented as the mean with standard error (SEM) or standard deviation (SD) of at least triplicates. Bar graph figures were generated using Graphpad Prism (version 8.0.1). The sample size ($n$) defines the number of mice for animal experiments and the number of replicates for cell studies. All datapoints are displayed in the plots when possible. Western blot experiments were indpendently repeated at least twice with similar results.

When comparing data with one variable factor, statistical significance was calculated by two-tailed Student's $t$test using EXCEL 2010, the exact $p$ values are labaled in the figure. For data with two variable factors, statistical significance was determined by two-way ANOVA with Bonferoni posttest using SigmaPlot (SPSS version 12.3; $*p < 0.05$, $**p < 0.01$, and $***p < 0.001$). Factor A is time, and factor B is genotype or drug treatment. For data involving time as the third variant, statistical significance was determined by two-way repeated measures ANOVA with Hold-Sidak posttest using SigmaPlot ($*p < 0.05$, $**p < 0.01$, and $***p < 0.001$). The genotype × timepoint interactions or treatment × timepoint interactions were tested and reported. Other statistical details of the experiments, such as the exact number of mice, can be found in the figure legends and in Supplementary Information files.

**Reporting summary**. Further information on research design is available in the Nature Research Reporting Summary linked to this article.

## Data availability
The structural models of COP1 and DET1 generated using I-Tasser are available upon request. All other data generated or analyzed during this study are included in this published article (and its Supplementary Information files). Data and reagents requests should be addressed to F.R. (raof@sustech.edu.cn). Source data are provided with this paper.

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

## Acknowledgements

This work was supported by grants from the National Science Foundation of China (31872798 and 91853129 to F.R.), the Shenzhen Municipal Government (KQJSCX20180322152418316, JCYJ20170412153517422, and JCYJ20170817104311912 to F.R.), the Department of Science and Technology of Guangdong Province (2018A030313207 to F.R.), and the Hong Kong Government GRF (17127718 and 17127215 to B.K.C.C.).

## Author contributions

Y.Y., Y.L., B.K.C.C., and F.R. designed research; H.L., Y.Y., Y.L., W.Y.S., X.Z., Xiaoli Yang, J.Z., X.W., Y.S., B.Z., and Xiuyan Yang performed research; B.K.C.C., W.H., and F.R. supervised research. N.J., K.Z., and F.W. contributed new reagents/analytic tools; H.L., Y.Y., X.Z., and F.R. analyzed data; and F.R. wrote the paper. All authors reviewed and edited the manuscript.

## Competing interests

The authors declare no competing interests.
