## [Peer Review File · Nature Communications]

Reviewers' Comments:

Reviewer #1:

Remarks to the Author:

The manuscript by Hong Lin et al. characterizes the previously generated and published mouse line carrying the CSN2 K70E mutant, which fails to interact with IP6. Whereas Csn2K70E/K70E embryos die at the stage of peri-implantation, Csn2WT/K70E heterozygous mice are born at the expected Mendelian ratio and display congenital hyperinsulinemia, insulin resistance and aggravated diet-induced obesity/diabetes, caused by increased secretion of insulin by pancreatic β cells. By performing an in-islet viral knockdown assay to screen for CUL4A substrate receptors that regulate insulin secretion, they identify COP1 as a prominent positive regulator of basal and glucose-stimulated islet insulin secretion. The CUL4-COP1-ETV5 axis is then identified as responsible for this effect, with ETV5 being the sole negative regulator of nutrient-induced insulin release among other Pea3 transcription factors.

The authors need to address some major unanswered questions and concerns prior to publication:

How do the insulin levels measured in the islets stimulated with glucose or other insulin secretagogues (KCl and L-Arginine) compare to basal levels? The authors need to include this data as well in their graph, or present the data as fold change compared to basal levels.

MLN4924 treatment leads to accumulation of ETV5 and restores serum insulin levels and body weight of CSN2WT/K70E mice. Since MLN4924 treatment in WT mice also leads to accumulation of ETV5 (as also seen in Figure 3I), how come that the authors do not see differences in serum insulin levels in case of WT treated with MLN4924?

RNAseq data will be instrumental in identifying genes which expression is altered upon Csn2 mutation in an unbiased manner, instead of focusing on Exoc6 and Syt13 genes, which have already been identified as direct ETVs targets.

Page 14: last sentence of the first paragraph needs to be rearranged/rewritten as it is unclear the point that the authors are trying to make.

In Figure 6A the authors show that treatment with MLN4924 in HFD mice leads to stabilization of ETV5. How do ETV5 levels change in NCD mice upon treatment with MLN4924? If ETV5 does indeed accumulate then it is difficult to reconcile with the observation that MLN4924 treatment does not affect insulin levels or insulin sensitivity under normal chow diet.

Other minor comments:

1. The authors need to add molecular weight markers to all of their western blots.
2. There are typos throughout the manuscript that the authors should correct (examples: Page 9, line 7: does instead of dose; Page 12, line 8: Fig 4A is in bold; Page 14, line 17: affect instead of affecting, etc).
3. Figure 1: labels on panels G and H are swapped. This also need to be corrected in the main text.
4. Figure 1K is missing the figure legend; it is labeled as (L) instead of (K).

Reviewer #2:

Remarks to the Author:

This study investigates the importance of the two complexes, IP6-CSN-CRL4 and CRL4-COP1-ETV5, in the regulation of insulin secretion and glucose homeostasis. To this end, the authors used a mouse model with mutation in the CSN2 subunit of the deneddylase CSN preventing the binding of IP6 (Heterozygous Csn2WT/K70E) and consequently the association and inactivation of CRL4 (Cul4A/4B) by CSN. They performed *in vivo* and *ex vivo* metabolic experiments (IPGTT, IPITT, insulin secretion). They also used HFD and ob/ob mice models to demonstrate the potential importance of CRL4-COP1-ETV5 axis in the regulation of obesity/type 2 diabetes. To evaluate the associations of these different complex partners they used molecular and pharmacological approaches (shRNA-islets-Ins1 beta cells; inhibitor of neddylation MLN4924). Finally, they studied nutritional and glucose regulation of these 2 complexes. They observed that heterozygous Csn2WT/K70E mice have improved glucose tolerance, are insulin resistant, hyperinsulinemic and heavier than WT mice. These mice have an increased glucose, KCL and arginine-stimulated insulin secretion, higher insulin content and Ins1 and Ins2 gene expression and an augmentation of insulin granules docking. In heterozygous Csn2WT/K70E, inhibition of CRL4 neddylation that prevents CRL4 activation normalizes insulinemia and body weight. In these mice, they showed that CRL4 neddylation favors the interaction to COP1/DET1 and the ubiquitination and degradation of the transcription factor ETV5 and decreased expression of two inhibitory exocytotic genes synaptotagmin-like 3 (Sytl3) and exocyst 6 (Exoc6). They also documented that COP1 and CSN compete for the binding to CRL4 and in the heterozygous Csn2WT/K70E the reduction of the formation of CSN-CRL4 complex favors the formation of a CRL4-COP1 complex. Increasing glucose concentration and refeeding rapidly diminished ETV5 protein levels that can be prevented by inhibition of CRL4 neddylation. By contrast, fasting and glucose deprivation increase ETV5 levels. In hyperglycemic hyperinsulinemic obese glucose intolerant and insulin resistant HFD and Ob/Ob mouse models, ETV5 level is low but when these animals were treated with the neddylation inhibitor, ETV5 level increased and all these parameters were improved.

General comments.

This study provides evidence for a regulatory axis implicated in the tight control of insulin secretion that involves 2 complexes: CSN-IP6-CRL4 which inhibits insulin secretion by preventing the formation of CRL4-COP1/DEP1-ETV5 complex and the degradation of ETV5. COP1-induced ETV5 degradation was previously implicated in the regulation of insulin secretion and the novelty of this study is the implication of CSN for the competition to CRL4 binding to COP1. The results with the two diabetic obese mouse models are particularly interesting related to the view proposed by the authors that specific inhibition of CRL4 may be a target to reduce hyperinsulinemia. Using the different mouse models, they convincingly show that hyperinsulinemia drives insulin resistance. Overall this work is interesting and well performed. It uses an extensive and diverse panel of experiments to unravel the different interactions in the regulatory CSN-IP6-CRL4-COP1/DEP1-ETV5 axis. However, some important clarifications should be made, and experiments should be performed not only in an unrelated cell line but in normal islets to prove the relevance and existence of both CSN-IP6-CRL4 and CRL4-COP1/DEP1-ETV5 complexes in normal beta cells. The importance of variations in IP6 levels in these processes and complexes should be clarified. Finally, some experiments should be performed in human islets or human beta cells to validate the relevance of the work and for drug development related to hyperinsulinemia, obesity and diabetes.

Major

- 1- The authors clearly demonstrated in total pancreas from fasted mice decreased association of Cul4A/B to COP1 (Fig 4F) and only report in HEK293 cells stably expressing myc-CSN2 that fasting increased CSN-Cul4A/ association (Fig 4G). These CSN-CRL4 and Cul4A/B-COP1 associations are particularly important in this study and should be proven in normal islets. The pancreas has 99% non-beta cells so how the author can conclude something about beta cells? If the problem is the amount of tissue, then it would be better to use for example INS832-13 cells.
- 2- In a previous study by this group (ref 25), the importance of IP6 for the sequestration of CRL4 by CSN was demonstrated. Also, despite having variations in ETV5 protein under diverse nutritional and glucose levels conditions that suggest modifications of the CSN-CRL4 and CRL4-

COP1 complexes, no change in IL6 levels was observed in the fast and fed conditions (sup fig 3 F and G). This questions the importance of IL6 in the regulation of the formation of these complex. To prove the importance of IP6, they should demonstrate that IP6 addition increases the CSN2-CRL4 association in WT islet but not or less in heterozygous Csn2WT/K70E mouse islets.

3- ETV5 was reported in a GWAS study as a gene associated with obesity/type 2 diabetes and COP1 auto-degradation was observed in T2D beta-cells. Thus, it would be important to show in human islets or EndoR-C1 human beta cell lines that CRL4-COP1-ETV5 is also important for the control of insulin secretion. The following keys experiments should be done: glucose stimulated insulin secretion +/- MLN4924 to prove the importance of CRL4 neddylation (as fig3A), dose dependence glucose effect on ETV5 expression (as fig 5E), and some molecular experiment using shCOP1 (as fig 3F) and shETV5 (as fig 3N) and measure glucose-stimulated insulin secretion.

Minor

1- The authors should give more details in the figure legends (Fig4F: fast duration; sup Fig 3E: what was the glucose concentration before glucose deprivation?) and better described experiments in the experimental procedure section (lentiviral shRNA-based knockdown screening for CRL4 adaptors; streptozocin mouse model).

2- Several Typo are present in the manuscript: in legend figures: fig1 (p28): hyperinsulinemia (line 3), glucose tolerance (line 8), secrete (line 16); sup fig 1 (p1): hyperinsulinemia (line 4); sup Fig 3 (p1): cellular (line 26); fig 4 (p29): immunoprecipitation (line 25); fig 7 (p30): sequestration (line 15); in figure 5C fasted; in main text: p8 line 16 immunofluorescence; p12 line 16 coimmunoprecipitation

3- In Fig 1B, heterozygous Csn2WT/K70E mice show fasting hypoglycemia (4 mM) in comparison to WT mice however in the same condition at time 0 min IPGTT (Fig 1E), heterozygous Csn2WT/K70E and WT mice have the same glycemia (5 mM). This difference should be explained.

4- Fig 1F and 1H, IPITT is expressed by % blood glucose so the author should provide time 0 glycemia after 4-6h fasting.

5- The information related to IP5K antibody and shRNAIP5K in experimental procedures should be removed.

6- In the sentence p7 line 5 'Compared to wildtype littermates, the Csn2WT/K70E mice have normal levels of blood glucose', fed should be added.

7- C57Bl6/J should be written C57Bl/6J (experimental procedures, p20).

8- In fig 4E legend, which lysate was used.

Reviewer #3:

Remarks to the Author:

Manuscript ID: Nature Communications manuscript NCOMMS-20-13724

Type of manuscript: Article

Title: IP6-assisted COP9-COP1 competition regulates a CRL4-ETV5 proteolytic checkpoint to safeguard glucose-induced insulin secretion against hyperinsulinemia

Authors: Hong Lin, Yuan Yan, Xiayun Wei, Xiaozhe Zhang, Xiaoli Yang, Jun Zhang, Yang Su, Yifan Luo, Xiuyan Yang, Fengchao Wang, and Feng Rao

The highly interesting manuscript by Lin et al. demonstrates the connection between glucose-induced insulin secretion and hyperinsulinemia on a molecular level. Moreover, the authors show an evolutionarily conserved interaction between constitutive photomorphogenesis 1 (COP1) and COP9 signalosome (CSN) from photomorphogenesis in plants to glucose homeostasis in mammals. Initially, the authors generate an excellent tool, heterozygous Csn2WT/K70E mice. The mutation of CSN2K70E leads to loss of IP6 binding and selective abolishment of IP6-augmented CSN-CRL interaction. Homozygous mice are not viable. Heterozygous mice are characterized by fasting hypoglycemia and hyperinsulinemia with enhanced insulin secretion. Adult Csn2WT/K70E mice

develop early signs of type II diabetes, including congenital hyperinsulinism, insulin resistance and obesity.

Using the inhibitor of neddylation, MLN4924, the authors found that a cullin-RING-ubiquitin ligase (CRL)-dependent process is responsible for the phenotype. Since CUL4A and CUL4B were highly neddylated in Csn2WT/K70E mice cells, they concluded that CRL4A/B complexes might be involved. COP1 was identified as a positive regulator of insulin secretion. Downregulation of COP1 and DET1 as well as MLN4924 revealed accumulation of a transcription factor, ETV5, that regulates insulin secretion. These data suggest that partial abolishment of CSN sequestration activates a CRL4-DET1-COP1-ETV5 degradation axis in Csn2WT/K70E pancreas. ETV5 regulates the secretion of insulin by its anti-exocytotic effects. The authors show that the CSN-CRL4COP1-ETV5 axis is a glucose-regulated checkpoint for nutrient-induced insulin secretion. ETV5 is degraded in a glucose-dependent manner to relieve the inhibition on insulin secretion, which is constitutively augmented in Csn2WT/K70E. Interestingly, CRL neddylation inhibition by MLN4928 prevents ETV5 degradation and hyperinsulinemia in diet- and leptin deficiency-induced obesity/diabetes.

In other words, the authors discovered that the competition between COP1 and IP6-CSN for binding to CUL4 regulates the stability of ETV5, which controls insulin secretion. Deregulated IP6-CSN-CRL4COP1-ETV5 axis causes hyperinsulinemia. However, it can be relieved by MLN4924 reducing obesity and diabetic risk, because of the correlation between insulin levels and obesity/diabetes in humans.

This opens new fields for diabetes and obesity treatment.

Minor points:

- The authors show co-immunoprecipitation of COP1 with CUL4A and CUL4B. Are CRL4ACOP1 and CRL4BCOP1 complexes equally involved in ETV5 ubiquitylation? Please make a statement in DISCUSSION.
- In Fig3D, it would be more proper to indicate the neddylated and deneddylated forms for all cullins.
- In Fig4D, it should be α -Cul4B but not α -Cul4A.

Reviewer #4:

Remarks to the Author:

Dr. Rao and colleagues in this article entitled "IP6-assisted COP9-COP1 competition regulates a CRL4-ETV5 proteolytic checkpoint to safeguard glucose-induced insulin secretion against hyperinsulinemia" have presented a study to elucidate the role of COP-1 in insulin secretion.

In Fig 4A the authors provide a structural model of how COP1 binds to CRL4, mediated by DET1. Both COP1 and DET1 was modeled using I-TASSER platform, however the authors have provided no other detail on modeling . For example, it would be important to know:

1. What were the UniProt ids of the sequence used.
2. Which templates (homologs) were used for modeling, what is their sequence identity to query protein.
3. What is the confidence of the models (C-score).
4. Can the authors ask Zhanglab to keep the models for reproducibility of results in future .
5. How did the authors fit DET1 in CRL4DDB2-CSN ? How similar is DET1 to DDB2.

Reviewer #1

The manuscript by Hong Lin et al. characterizes the previously generated and published mouse line carrying the CSN2 K70E mutant, which fails to interact with IP6. Whereas Csn2K70E/K70E embryos die at the stage of peri-implantation, Csn2WT/K70E heterozygous mice are born at the expected Mendelian ratio and display congenital hyperinsulinemia, insulin resistance and aggravated diet-induced obesity/diabetes, caused by increased secretion of insulin by pancreatic β cells. By performing an in-islet viral knockdown assay to screen for CUL4A substrate receptors that regulate insulin secretion, they identify COP1 as a prominent positive regulator of basal and glucose-stimulated islet insulin secretion. The CUL4-COP1-ETV5 axis is then identified as responsible for this effect, with ETV5 being the sole negative regulator of nutrient-induced insulin release among other Pea3 transcription factors. The authors need to address some major unanswered questions and concerns prior to publication:

1. How do the insulin levels measured in the islets stimulated with glucose or other insulin secretagogues (KCl and L-Arginine) compare to basal levels? The authors need to include this data as well in their graph, or present the data as fold change compared to basal levels.

In figure 1K of the original manuscript, islet insulin secretion was plotted as relative values normalized to WT-basal levels. We have now included in Fig 1K a plot of fold change by comparing “stimulated” VS “basal” for WT or Het islets, which still showed significantly higher fold changes upon glucose and L-arginine stimulation. Nonetheless, the key difference is constitutively higher insulin secretion in both basal and stimulated states for Het islets.

2. MLN4924 treatment leads to accumulation of ETV5 and restores serum insulin levels and body weight of CSN2WT/K70E mice. Since MLN4924 treatment in WT mice also leads to accumulation of ETV5 (as also seen in Figure 3I), how come that the authors do not see differences in serum insulin levels in case of WT treated with MLN4924?

We are glad to receive this question from the reviewer, which really makes glycemic regulation of ETV5 more interesting. First of all, please note that Fig 3I is not an experiment on mice. Rather, Fig 3I examined islets cultured in 11 mM glucose, a high-glucose condition whereby ETV5 is expected to be degraded by CUL4^{COP1}, and thus stabilized upon MLN4924-mediated CUL4 inhibition.

Physiologic ETV5 degradation depends on the degree of glycemia. Thus, ETV5 is excessively degraded in hyperglycemic HFD-fed and ob/ob mice (Fig 6A, 6E). Under normal, free-feeding condition, we found no significant effect of MLN4924 on steady-state ETV5 levels in WT mice (revised Fig S5A), unless we perform a (overnight) fasting-then-refeeding (1 h) experiment to enlarge the difference (Fig 5G). Without prominent effect on ETV5, it is not surprising that MLN4924 has no major influence on insulin levels in WT mice. In CSN2^{WT/K70E} mice, pancreatic ETV5 is

excessively degraded due to constitutively augmented assembly of CRL4^{COP1} E3 ligase. MLN4924 then could have a prominent effect to curb constitutive insulin secretion.

The new data in revised FigS5A is now described on page 16, highlighted in red. We also discuss glycemia-stimulated ETV5 degradation by CRL4^{COP1} on page 17, first paragraph, highlighted in red.

3. RNAseq data will be instrumental in identifying genes which expression is altered upon Csn2 mutation in an unbiased manner, instead of focusing on *Exoc6* and *Sytl3* genes, which have already been identified as direct ETVs targets.

We appreciate this question from the reviewer. The main message of our study is glucose-dependent transition of CRL4 from CSN-bound, inactive state to COP1-bound, active state during insulin secretion, with the IP₆ small molecule functioning as a determinant between these two states. We have also identified ETV5 degradation as the main downstream event of glucose-induced CRL4^{COP1} activation. To further support these core findings, we examined and found congruent transcriptional changes of ETV targets *Sytl3* and *Exoc6*, which are insulin exocytosis regulators and were previously identified by a reputed lab. We did not initially aim to identify new ETV targets, which falls beyond the expertise of our lab. During this revision, we have attempted the suggested RNA-seq experiments. However, as seen from the PCA plot shown below, the data quality seems rather poor, with WT and Het islet triplicates being scattered all over the plot. We decided not to pursue this further.

4. Page 14: last sentence of the first paragraph needs to be rearranged/rewritten as it is unclear the point that the authors are trying to make.

We have rewritten this sentence in a more concise manner, now on page 14 the last paragraph and highlighted in red.

5. In Figure 6A the authors show that treatment with MLN4924 in HFD mice leads to stabilization of ETV5. How do ETV5 levels change in NCD mice upon treatment with MLN4924? If ETV5 does indeed accumulate then it is difficult to reconcile with the observation that MLN4924 treatment does not affect insulin levels or insulin sensitivity under normal chow diet.

The effect of MLN4924 on ETV5 levels in NCD mice is also part of comment #2 from this reviewer. We found no significant effect of MLN4924 on steady-state ETV5 levels under normal feeding conditions. This new data is presented in Fig S5A, and is described on page 16, highlighted in red. Consistent with this data, we did not observe statistically significant effect of MLN4924 on insulin levels for NCD mice (Fig 6C).

Other minor comments:

1. The authors need to add molecular weight markers to all of their western blots.
We have added molecular weight markers to all of our western blots.
2. There are typos throughout the manuscript that the authors should correct (examples: Page 9, line 7: does instead of dose; Page 12, line 8: Fig 4A is in bold; Page 14, line 17: affect instead of affecting, etc).
3. Figure 1: labels on panels G and H are swapped. This also need to be corrected in the main text.
4. Figure 1K is missing the figure legend; it is labeled as (L) instead of (K).
2-4: We have corrected the typos, panel labels and figure legends. We genuinely appreciate careful reading of the manuscript from this reviewer.

Reviewer #2:

General comments.

This study provides evidence for a regulatory axis implicated in the tight control of insulin secretion that involves 2 complexes: CSN-IP6-CRL4 which inhibits insulin secretion by preventing the formation of CRL4-COP1/DEP1-ETV5 complex and the degradation of ETV5. COP1-induced ETV5 degradation was previously implicated in the regulation of insulin secretion and the novelty of this study is the implication of CSN for the competition to CRL4 binding to COP1. The results with the two diabetic obese mouse models are particularly interesting related to the view proposed by the authors that specific inhibition of CRL4 may be a target to reduce hyperinsulinemia. Using the different mouse models, they convincingly show that hyperinsulinemia drives insulin resistance. Overall this work is interesting and well performed. It uses an extensive and diverse panel of experiments to unravel the different interactions in the regulatory CSN-IP6-CRL4-COP1/DEP1-ETV5 axis. However, some important clarifications should be made, and experiments should be performed not only in an unrelated cell line but in normal islets to prove the relevance and existence of both CSN-IP6-CRL4 and CRL4-COP1/DEP1-ETV5 complexes in normal beta cells. The

importance of variations in IP6 levels in these processes and complexes should be clarified. Finally, some experiments should be performed in human islets or human beta cells to validate the relevance of the work and for drug development related to hyperinsulinemia, obesity and diabetes.

Major

1- The authors clearly demonstrated in total pancreas from fasted mice decreased association of Cul4A/B to COP1 (Fig 4F) and only report in HEK293 cells stably expressing myc-CSN2 that fasting increased CSN-Cul4A/ association (Fig 4G). These CSN-CRL4 and Cul4A/B-COP1 associations are particularly important in this study and should be proven in normal islets. The pancreas has 99% non-beta cells so how the author can conclude something about beta cells? If the problem is the amount of tissue, then it would be better to use for example INS832-13 cells.

We agree with this reviewer that demonstrating CSN-CRL4 and Cul4A/B-COP1 complex competition in beta cells or islets is of particularly importance to this study. Because isolated islets do not yield sufficient amount of proteins for co-immunoprecipitation experiments, we employed the mouse β -cell line MIN6, in which we can successfully immunoprecipitate both Cul4B and CSN3 (we cannot immunoprecipitates Cul4B from the rat β -cell line INS1-832/13).

In revised Fig 4B, we show that COP1 and DET1 co-expression inhibits CSN-Cul4 interaction in MIN6 cells, same as originally observed in HEK293 cells. In revised Fig 4F, we show that fasting (-glucose) increased CSN-CRL4 association in MIN6 cells, which can be reversed by refeeding (+ glucose). In contrast, as shown in revised Figure 4G, Cul4B-COP1 association in MIN6 cells is decreased by fasting (-glucose), which can also be rescued by refeeding (+ glucose). These data have therefore verified that the proposed COP1-CSN competition for CRL4 works not only in HEK293 cells and pancreas, but also in β cell.

2- In a previous study by this group (ref 25), the importance of IP6 for the sequestration of CRL4 by CSN was demonstrated. Also, despite having variations in ETV5 protein under diverse nutritional and glucose levels conditions that suggest modifications of the CSN-CRL4 and CRL4-COP1 complexes, no change in IP6 levels was observed in the fast and fed conditions (sup fig 3 F and G). This questions the importance of IP6 in the regulation of the formation of these complex. To prove the importance of IP6, they should demonstrate that IP6 addition increases the CSN2-CRL4 association in WT islet but not or less in heterozygous Csn2WT/K70E mouse islets.

The lack of changes in IP6 levels under fasting and fed conditions could be because the changes are local, and the currently available method only can measure total cellular IP6 levels. Alternatively, glucose might work through other mechanisms. To avoid distraction, we removed this data from the current manuscript. Nonetheless, we concur with this reviewer that, on top of citing our previous work (ref 25), the importance of IP6 in vivo should be demonstrated by comparing wt and Csn2WT/K70E mice. Considering material limitation, we performed such an

experiment using pancreas instead of islet.

On page 9, we describe new data presented in Fig S2A showing that CSN-CRL4 interaction is indeed reduced in the CSN2^{WT/K70E} mice pancreas, which validates the purpose of generating this mutant. Furthermore, IP6 addition increased CSN-CRL4 interaction in both wildtype and CSN2^{WT/K70E} mice. This is expected, since the remaining wildtype allele in CSN2^{WT/K70E} mice still can respond to the bridging effect of IP6. In revised Fig 4E, we further show that IP6 addition dramatically reduces CRL4-COP1 association in beta cells, consistent with IP6 reciprocally regulating COP1-CSN competition for CRL4. As such, IP₆ could indeed be the limiting factor.

3- ETV5 was reported in a GWAS study as a gene associated with obesity/type 2 diabetes and COP1 auto-degradation was observed in T2D beta-cells. Thus, it would be important to show in human islets or EndoR-C1 human beta cell lines that CRL4-COP1-ETV5 is also important for the control of insulin secretion. The following key experiments should be done: glucose stimulated insulin secretion +/- MLN4924 to prove the importance of CRL4 neddylation (as fig3A), dose dependence glucose effect on ETV5 expression (as fig 5E), and some molecular experiment using shCOP1 (as fig 3F) and shETV5 (as fig 3N) and measure glucose-stimulated insulin secretion.

We have obtained data using both human islets and the EndoC-βH1 cells, which suggests that the conclusions we draw in mice also apply to human.

Through collaboration with clinician scientist from the 2nd affiliated Hospital of our University, we studied the effect of CRL inhibition (MLN4924) and COP1/ETV5 knockdown on insulin secretion from human islet (Fig 6A-C). MLN4924 and shCOP1 both inhibits insulin secretion, with more pronounced effects under high glucose conditions, when CRL4^{COP1} is more active. On the other hand, shETV5 increases insulin secretion from human islet under both low and high glucose conditions. We further collected enough human islets to perform ETV5 western blot under different glucose conditions, ETV5 is clearly degraded upon glucose stimulation (a non-specific band, marked by an asterisk, remain unchanged) (Fig 6D). These data from human islets are consistent with the conclusions we draw using mouse samples.

The human beta cell line EndoC-βH1 is not easily accessible. Through collaboration with Dr. Weiping Han's lab at Singapore Bioimaging Consortium, we also studied the effects of CRL inhibition (MLN4924) and COP1/ETV5 knockdown on insulin secretion from EndoC-βH1 cells (Fig 6E-G). Although EndoC-βH1 is not highly responsive in GSIS assays, MLN4924 and shCOP1 both inhibits insulin secretion, with more pronounced effects under high glucose conditions, when CRL4^{COP1} is more active. ShETV5 increases insulin secretion under both low and high glucose conditions. Moreover, ETV5 is degraded upon increasing glucose stimulation in a dose-dependent manner (Fig 6H). These data from human β cells again supports the conclusions we draw using mouse samples.

The above data are now presented as new Fig 6 and described on page 15, highlighted in red.

Minor

1- The authors should give more details in the figure legends (Fig4F: fast duration; sup Fig 3E: what was the glucose concentration before glucose deprivation?) and better described experiments in the experimental procedure section (lentiviral shRNA-based knockdown screening for CRL4 adaptors; streptozocin mouse model).

We have added the requested details to Fig 4F and sup Fig 3E. We've also described the procedures for Streptozocin treatment and shRNA-based knockdown screening for CRL4 adaptors in the methods section on page 22 and 24, highlighted in red.

2- Several Typos are present in the manuscript: in legend figures: fig1 (p28): hyperinsulinemia (line 3), glucose tolerance (line 8), secrete (line 16); sup fig 1 (p1): hyperinsulinemia (line 4); sup Fig 3 (p1): cellular (line 26); fig 4 (p29): immunoprecipitation (line 25); fig 7 (p30): sequestration (line 15); in figure 5C fasted; in main text: p8 line 16 immunofluorescence; p12 line 16 coimmunoprecipitation

We have corrected all the typos. We are thankful to this reviewer for the careful reading.

3- In Fig 1B, heterozygous Csn2^{WT/K70E} mice show fasting hypoglycemia (4 mM) in comparison to WT mice however in the same condition at time 0 min IPGTT (Fig 1E), heterozygous Csn2^{WT/K70E} and WT mice have the same glycemia (5 mM). This difference should be explained.

We are grateful to this reviewer for pointing out the differences in fasting glucose levels between Fig 1B and 1E. This is because data for Fig1B was obtained when the mice were between 2-2.5 months old, whereas mice were 4 months old by the time the IPGTT experiment in Fig1E was performed. As shown in the original Fig 1F (now Fig S1D), severe insulin resistance is observed in CSN2^{WT/K70E} mice at 4 month age, which would prevent fasting hypoglycemia.

We have re-performed the IPGTT experiments with a new set of WT and CSN2^{WT/K70E} littermates at 2.5 month age. As shown in revised Fig 1E, the CSN2^{WT/K70E} mice display fasting hypoglycemia and better glucose tolerance. An ITT experiment also showed significant insulin resistance (revised Fig 1F), which is however less severe than 4-month old mice (now Fig S1D). The age-dependent worsening of insulin sensitivity is in line with insulin resistance caused by chronic insulin over-secretion.

4- Fig 1F and 1H, IPITT is expressed by % blood glucose so the author should provide time 0 glycemia after 4-6h fasting.

As a common practice the ITT data were presented as % blood glucose and time 0 glycemia was not our focus, because, unlike GTT where mice were fasted for 16 h, ITT mice were fasted for 4-6 h, at which time point there is no significant glycemia difference between WT and CSN2^{WT/K70E} mice.

5- The information related to IP5K antibody and shRNAIP5K in experimental procedures should be removed.

This error has been corrected.

6- In the sentence p7 line 5 ‘Compared to wildtype littermates, the Csn2WT/K70E mice have normal levels of blood glucose’, fed should be added.

“fed” has been added and highlighted in red.

7- C57Bl6/J should be written C57Bl/6J (experimental procedures, p20).

This error has been corrected.

8- In fig 4E legend, which lysate was used.

The original Fig 4E experiment was performed using HEK293 cell lysates. In view of this reviewer’s suggestion to use beta-cells (major comment #1), we have replaced with new Fig 4E whereby lysates from the mouse β -cell line MIN6 were used to examine the effect of IP6 addition, consistent with the data obtained with HEK293 cell lysates, IP6 addition dramatically diminishes CRL4-COP1 interaction in MIN6 cell lysates, suggesting that IP6 regulation of CRL4^{COP1} assembly is a fundamental biochemical phenomenon potentially conserved across species and tissue type. The Figure legend for Fig 4E is now updated.

Reviewer #3:

The highly interesting manuscript by Lin et al. demonstrates the connection between glucose-induced insulin secretion and hyperinsulinemia on a molecular level. Moreover, the authors show an evolutionarily conserved interaction between constitutive photomorphogenesis 1 (COP1) and COP9 signalosome (CSN) from photomorphogenesis in plants to glucose homeostasis in mammals.

Initially, the authors generate an excellent tool, heterozygous Csn2WT/K70E mice. The mutation of CSN2K70E leads to loss of IP6 binding and selective abolishment of IP6-augmented CSN-CRL interaction. Homozygous mice are not viable. Heterozygous mice are characterized by fasting hypoglycemia and hyperinsulinemia with enhanced insulin secretion. Adult Csn2WT/K70E mice develop early signs of

type II diabetes, including congenital hyperinsulinism, insulin resistance and obesity. Using the inhibitor of neddylation, MLN4924, the authors found that a cullin-RING-ubiquitin ligase (CRL)-dependent process is responsible for the phenotype. Since CUL4A and CUL4B were highly neddylated in Csn2WT/K70E mice cells, they concluded that CRL4A/B complexes might be involved. COP1 was identified as a positive regulator of insulin secretion. Downregulation of COP1 and DET1 as well as MLN4924 revealed accumulation of a transcription factor, ETV5, that regulates insulin secretion. These data suggest that partial abolishment of CSN sequestration activates a CRL4-DET1-COP1-ETV5 degradation axis in Csn2WT/K70E pancreas. ETV5 regulates the secretion of insulin by its anti-exocytotic effects. The authors show that the CSN-CRL4COP1-ETV5 axis is a glucose-regulated checkpoint for nutrient-induced insulin secretion. ETV5 is degraded in a glucose-dependent manner to relieve the inhibition on insulin secretion, which is constitutively augmented in Csn2WT/K70E. Interestingly, CRL neddylation inhibition by MLN4928 prevents ETV5 degradation and hyperinsulinemia in diet- and leptin deficiency-induced obesity/diabetes.

In other words, the authors discovered that the competition between COP1 and IP6-CSN for binding to CUL4 regulates the stability of ETV5, which controls insulin secretion. Deregulated IP6-CSN-CRL4COP1-ETV5 axis causes hyperinsulinemia. However, it can be relieved by MLN4924 reducing obesity and diabetic risk, because of the correlation between insulin levels and obesity/diabetes in humans.

This opens new fields for diabetes and obesity treatment.

Minor points:

- The authors show co-immunoprecipitation of COP1 with CUL4A and CUL4B. Are CRL4ACOP1 and CRL4BCOP1 complexes equally involved in ETV5 ubiquitylation? Please make a statement in DISCUSSION.

We thank this reviewer for kindly reminding us to discuss Cul4A and Cul4B. Indeed, given that the neddylation levels of Cul4A and Cul4B (Fig 3D), and the assembly of CRL4A^{COP1} and CRL4B^{COP1} E3 ligases (Fig 4C-D), are both increased in CSN2^{WT/K70E} mice, it remains uncertain if CRL4A^{COP1} and CRL4B^{COP1} are equally involved in ETV5 ubiquitylation. We have now discussed this point on page 18, highlighted in red.

- In Fig3D, it would be more proper to indicate the neddylated and deneddylated forms for all cullins.

The neddylated and deneddylated forms of all cullins in Fig3D have been indicated.

- In Fig4D, it should be α -Cul4B but not α -Cul4A.

This error has been corrected.

Reviewer #4 (Remarks to the Author):

Dr. Rao and colleagues in this article entitled "IP6-assisted COP9-COP1 competition regulates a CRL4-ETV5 proteolytic checkpoint to safeguard glucose-induced insulin secretion against hyperinsulinemia" have presented a study to elucidate the role of COP-1 in insulin secretion.

In Fig 4A the authors provide a structural model of how COP1 binds to CRL4, mediated by DET1. Both COP1 and DET1 was modeled using I-TASSER platform, however the authors have provided no other detail on modeling. For example, it would be important to know:

We thank this reviewer for bring up questions regarding the modeling part of the work. Before addressing the specific questions from this reviewer, we would like to convey the following message:

Our modeling was actually based on a combination of prior literature search and our own unpublished data, which are now presented in Figure S3 and described on page 12. These data and their use in guiding the modeling process are described below:

1st, prior work has shown that DET1, a DCAF, mediates CRL4 interaction with COP1 (ref 44, 45), which we verified by DET1 knockdown experiments (Fig S3A).

2nd, the N-terminal 100 amino acid of DET1 is homologous to other DCAFs and contains the characteristic alpha helical motif that critically binds DDB1 (Nat. Struc. Mol. Biol. 2010 17: 105-111). Consistently, our fragment mapping studies suggest that the N-terminal region of DET1 binds DDB1 (Fig S3C). Aligning the alpha helical motif of DET1 to its counterpart in DDB2 is an essential step in building the model of CRL4^{DET1/COP1}.

3rd, prior work has suggested that COP1 interaction with DET1 requires its exon 7 (aa 277-296) (ref 45), which we verified by demonstrating that the Δ exon7 mutant COP1 fails to interact with DET1 (Fig S3B). Moreover, the Δ exon7 mutant COP1 fails to interact with DDB1, consistently with this being indirect and via DET1. This COP1-DET1 binding interface information is used in building the final model.

4th, we further identified the region of DET1 that binds COP1 via reciprocal immunoprecipitation. The data suggest that the very C-terminal portion of DET1 (aa 451-550) specifically binds COP1, whereas aa 287-450 is non-specifically pulled down by HA-beads (Fig S3D-E). This COP1-DET1 binding interface information is also used to position COP1 against DET1 in the final model.

It is with the above experimental work that we build the final model of CRL4^{DET1/COP1}. We have now described this modeling procedure in greater detail on page 27, highlighted in red.

In the cryo-EM structure of CRL4^{DDB2}-CSN, two CSN subunits, CSN1 and CSN3, are very close to DDB2 on both sides. It is therefore within the expectation that, while DET1 replacement of DDB2 can be accommodated, additional placement of another large protein like COP1, or a ubiquitylation substrate, would result in steric clash with CSN1, CSN3, or both.

Lastly, the model in Fig 4A is merely to visually illustrate potential CSN-COP1 completion. More importantly, we provided ample experimental data supporting

COP-CSN1 competition in Fig 4B-G and Fig S1A. For example, we showed that overexpressing COP1/DET1 inhibits CSN-CRL interaction (Fig 4B), that glucose availability reciprocally regulates CRL4-CSN VS CRL4-COP1 association (Fig 4F-G), and that IP6 addition reciprocally regulates CRL4-CSN VS CRL4-COP1 association (Fig 4E, S1A). Therefore, our conclusions remain the same even without Fig 4A.

1. What were the UniProt ids of the sequence used.

This uniprot IDs (Q7L5Y6 for hDET1, Q8NHY2 for hCOP1) has been added into the method section.

2. Which templates (homologs) were used for modeling, what is their sequence identity to query protein.

DET1 and DDB1 are both >50 kDa and are multi-domain proteins. The I-Tasser program used different templates for different regions, and, for low confidence regions, many templates for the same region to build a consensus model.

For DET1, its N-terminal 100 amino acids contains a characteristic helical motif that critically binds DDB1 (Nat. Struc. Mol. Biol. 2010 17: 105-111), and was modeled based on DCAF15 from the structure of the DDB1-DDA1-DCAF complex (PDB id: 6PAI), which has sequence identity of 27%. The rest of the DET1 protein does not have a clear homolog, and the structural model was collectively built by using many templates in the threading program.

For COP1, crystal structures of its C-terminal WD40 domain has been solved (aa 376-731; PDB id: 5HQG) and is included in the final model. The N-terminal portion of COP1 does not have a clear homolog, and the structural model was collectively built by using many templates in the threading program.

3. What is the confidence of the models (C-score).

The C-score for the best DET1 model is -1.47; The C-score for the best COP1 model is -0.7;

4. Can the authors ask Zhanglab to keep the models for reproducibility of results in future .

I-Tasser is widely regarded as one of the most reliable homology modeling server, repeatedly performing the best in CASP (Critical Assessment of Structure Prediction) tests. We cannot ask anything from the I-Tasser authors. However, we have recently re-uploaded COP1 and DET1 for homology modeling. The outcome can be access at: <https://zhanglab.ccmb.med.umich.edu/I-TASSER/output/S567614/> for COP1 <https://zhanglab.ccmb.med.umich.edu/I-TASSER/output/S567620/> for DET1.

5. How did the authors fit DET1 in CRL4DDB2-CSN ? How similar is DET1 to DDB2.

Because all DCAFs, including DDB2 and DET1, employs a characteristic helical motif to bind the same groove in DDB1 (Nat. Struc. Mol. Biol. 2010 17: 105-111),

and because we showed that the same helical motif in DET is required for DDB1 binding (Fig S3C), DET1 was therefore fitted by aligning its N-terminal helical motif with that of DDB2.

We would like to thank you and the reviewers for the time and invaluable insights, which help to significantly improve and strengthen our current work. We hope that you will find this letter and the edited manuscript address the reviewer's concerns.

On behalf of all authors,

Feng Rao

Reviewers' Comments:

Reviewer #1:

Remarks to the Author:

Concerns have been adequately addressed.

Reviewer #3:

Remarks to the Author:

Manuscript ID: Revised Nature Communications manuscript NCOMMS-20-13724A

Type of manuscript: Article

Title: IP6-assisted COP9-COP1 competition regulates a CRL4-ETV5 proteolytic checkpoint to safeguard glucose-induced insulin secretion against hyperinsulinemia

Authors: Hong Lin, Yuan Yan, Yifan Luo, Wing Yan So, Xiayun Wei, Xiaozhe Zhang, Xiaoli Yang, Jun Zhang, Yang Su, Xiuyan Yang, Bobo Zhang, Kangjun Zhang, Nan Jiang, Billy Kwok Chong Chow, Weiping Han, Fengchao Wang and Feng Rao

All questions raised by the reviewer were addressed perfectly. The revision improved and completed the manuscript.

Reviewer #4:

Remarks to the Author:

Thanks to the authors for systematically addressing the questions and incorporating it in the manuscript.

I would highly recommend the authors to download the I-TASSER results for both hDET1 and hCOP1 (download links available on the top of the result page) and submit the downloaded zip files as online supplementary material. Since the template library of I-TASSER keeps changing on a weekly basis, reproducing exact same modeling results (by re-running I-TASSER even on same sequence) would be impossible.

Reviewer #5:

Remarks to the Author:

I did not see the original paper and was requested to comment on whether the comments made by Reviewer 2 have been satisfactorily addressed.

I agree with the original Reviewer II that this is an interested and well-performed study.

Reviewer 2 requested (1) clarification on the 'importance of variations in IP6 levels' and (2) that key experiments are repeated in human islets/beta-cells.

The authors have satisfactorily dealt with (2) and decided to omit (1).

Although I should - in fairness to the authors - refrain from raising additional comments at this stage, I nevertheless take the liberty of highlighting a few aspects that in my opinion requires further attention (and may make the manuscript even stronger).

1) The way the authors express their secretion data worries me. Why normalise to basal in wildtype (are they just dividing secretion data with the mean at low glucose? - how is this even possible statistically).

It would be much better to express data as ng per islet and/or relate it to insulin content (here and everywhere else).

I am NOT at all persuaded that there is a true effect on glucose-induced insulin secretion. A simple explanation for all their data is that secretion is related to insulin content and that the Csn2wt/k70e mice simply secrete more insulin because they contain more insulin!

2) Fig. 2F. An insulin content of 1 ng/islet is unbelievably low. Mouse islets typically contain 50-100 ng/islet! Please check calculations!

3) It is good that the number of docked granules have been measured (Fig. 2H). What about global granule density. Is it also increased by 100%

4) Page 10, line 13: 'eliminates the difference...'. I think this statement is rather misleading. It not only eliminates the difference, it in fact eliminates insulin secretion altogether! The islets might not be doing very well and this results is not credible.

5) To refer to incubation of cells at different glucose (p. 14, lines 2. as 'feeding' is stretching it a bit....

6) Page 15. Human islet data. How many human islet preparations were used? It is good practice to present donor characteristics in a supplementary table (sex, bmi, age)

7) If blood glucose concentrations in Fig. 1F-G (and elsewhere) are to be quoted as % (of t=0), then the legend should contain the information on how much 100% is! The effect on insulin tolerance in Fig. 1F is best described as 'marginal'.

8) Fig. 1J: y-axis (normalised to basal?)

9) Fig. 3A. Were islets pretreated for 2h with MLN or were experiments conducted in the continued presence of the compound? The effects are remarkably strong (>90%) even in WT islets and yet there is no difference on serum insulin? There is a glaring discrepancy between the data in fig. 3A-B! I would feel more comfortable if the authors could show that MLN is not simply poisoning the islets! (what happens if MLN is applied in the calcium measurements?)

10) Statistical analyses. The authors clearly like Student's t-test and use it widely - also when it may not be appropriate. Given that many of the effects they report are relatively small, it is important that the appropriate statistical tests are used to bolster their conclusions.

Reviewer #4 (Remarks to the Author):

Thanks to the authors for systematically addressing the questions and incorporating it in the manuscript.

I would highly recommend the authors to download the I-TASSER results for both hDET1 and hCOP1 (download links available on the top of the result page) and submit the downloaded zip files as online supplementary material. Since the template library of I-TASSER keeps changing on a weekly basis, reproducing exact same modeling results (by re-running I-TASSER even on same sequence) would be impossible.

A: We appreciate the comment on the frequent updates in I-TASSER library. As suggested, we have included the I-TASSER results as online supplementary material in the current revised version. For hDET1 and hCOP1, the I-TASSER models before and after manuscript submission are very similar.

Reviewer #5 (Remarks to the Author):

General comments.

I did not see the original paper and was requested to comment on whether the comments made by Reviewer 2 have been satisfactorily addressed. I agree with the original Reviewer II that this is an interested and well-performed study.

A: We appreciate the favorable comment from reviewer #5.

Reviewer 2 requested (1) clarification on the 'importance of variations in IP6 levels' and (2) that key experiments are repeated in human islets/beta-cells.

The authors have satisfactorily dealt with (2) and decided to omit (1).

A: In the previous revised version we provided new data and analyses to address the request (1), but the writing might be unclear. We have now revised the text and highlighted the changes that are relevant to the request (1) in the current revised version.

In his/her original comments, Reviewer 2 acknowledged our efforts to examine whether levels of IP6 change during fasting-induced CRL4^{COP1} transition into CRL-CSN. This reviewer further suggested that, despite the data showing no changes in total pancreatic IP6 levels (original Fig S3F), we could “prove the importance of IP6” by “demonstrating that IP6 addition increases the CSN2-CRL4 association in WT islet but not or less in heterozygous Csn2^{WT/K70E} mouse islets.”

In response to this request of reviewer #2, we performed new experiments and presented the results in Fig S2A, showing that endogenous CSN-CRL4 interaction is indeed reduced in the CSN2^{WT/K70E} mice pancreas. Furthermore, IP6 addition increased CSN-CRL4 interaction in wildtype mice and, to a lesser extent, in CSN2^{WT/K70E} mice. These data validates the importance of IP6 as a CSN2 cofactor. In the revise Fig 4E, we further showed that IP6 addition dramatically reduces CRL4-COP1 association in beta cells, suggesting that IP6 reciprocally regulates COP1-CSN competition for CRL4. As such, IP₆ could

indeed be the gating/limiting factor.

As discussed in our previous revision letter, due to technical limitations, current methods only detect total IP6 levels, but not changes of any individual local pool. Since the central message of Fig5, the main figure for the original Fig S3F, was to describe that glucose stimulation of $CRL4^{COP1}$ leads to degradation of ETV5, we did not include total IP6 levels in the previous revised version. We have now added the results showing the lack of changes in IP6 levels under fast-feeding transitions as new Fig S4G in the current revised version.

Specific Comments.

1) The way the authors express their secretion data worries me. Why normalise to basal in wildtype (are they just dividing secretion data with the mean at low glucose? - how is this even possible statistically). It would be much better to express data as ng per islet and/or relate it to insulin content (here and everywhere else).

A: Thanks for the comments. We agree that insulin secretion data are typically expressed as fractions of insulin content or normalized to islet amount. However, there are reports that use basal level for normalization, for example a particularly relevant report by Suriben et al (Cell, 2015, 163, 1457-1467: Fig 3B, 3D, 5A and 7C). We used islets of same size for one experiment (but not across experiments). The secretion data were then normalized to basal in wildtype by factoring the average of WT basal secretion as 1, with all the other datapoints dividing the same denominator.

For the current revised version, we have performed new islet experiments and used fractions of insulin content to express the secretion data (Fig 2I). As we did not measure total islet insulin content in previous experiments, we would not be able to present those data in the same manner, i.e. as fractions of insulin content. We could, however, present the data as ng/islet. But due to the islet size variations, we would only be able to compare within a specific experiment, but not across experiments.

I am NOT at all persuaded that there is a true effect on glucose-induced insulin secretion. A simple explanation for all their data is that secretion is related to insulin content and that the $Csn2wt/k70e$ mice simply secrete more insulin because they contain more insulin!

A: We thank the reviewer for this critical comment. We have performed new islet secretion experiments, where we could rule out the potential contribution of insulin content differences and directly assess exocytotic capacity. As shown in Fig 2I, Heterozygous (Het) islets secrete higher percentages of insulin content under both low and high glucose conditions, suggesting a true effect on secretion per se. Moreover, while WT islets secrete 16-fold more insulin on high glucose, Het islets secrete 24-fold more, suggesting that they are also more responsive to GSIS. This new data is described and discussed on page 8-9, highlighted in red in the current revised version.

We would like to emphasize that the above secretion% data is in line with many other data we've presented throughout the manuscript. Thus, known insulin exocytosis regulators, such as Exoc6 and Syt13, are differentially expressed in Het islets (Fig2G). Moreover, Exoc6 and Syt13 over-expression, especially their co-

expression, regulates insulin secretion (Fig 3P). In Figure 1K (right panel), WT islet showed about 4-fold increase in insulin secretion upon high glucose, whereas Het islets showed a 6.2 fold increase, again consistent with Het islets being more glucose-responsive. On the other hand, while MLN reduces basal insulin secretion by just 2-fold, it inhibits GSIS by 10-fold (WT islets) or 17-fold (Het islets) (Fig 3A), consistent with an important role of CRL disinhibition during GSIS.

COP1, a major downstream target of CSN identified in this work, has been noted to dysfunction in human islets with exocytotic defect (Camunas-Soler et al, Cell Metab.2020, 31, 1017-1031). As discussed below in addressing point #4 of this reviewer, we also found that COP1 knockdown diminishes basal insulin secretion and, more strikingly, GSIS. In view of the new and old data and prior literature, we conclude that Csn2wt/k70e mice secrete more insulin because it contains more insulin and also because it has higher exocytotic capability.

2) Fig. 2F. An insulin content of 1 ng/islet is unbelievably low. Mouse islets typically contain 50-100 ng/islet! Please check calculations!

A: We are grateful to this reviewer for pointing this out. We overlooked the dilution factor during calculation. We have now updated Fig 2F in the current revised version.

3) It is good that the number of docked granules have been measured (Fig. 2H). What about global granule density. Is it also increased by 100%.

A: This is an important question, but technically demanding. We carried out additional experiments to examine global granule density by taking EM images at lower (3000x) magnification. However, as illustrated below with two images of WT islets, granule density varies widely depending on whether the nucleus is included and possibly sample cutting direction. This leads us to conclude that future 3-D reconstruction or tissue-clearance methods are required to reliably determine global granule density.

4) Page 10, line 13: 'eliminates the difference...'. I think this statement is rather misleading. It not only eliminates the difference, it in fact eliminates insulin secretion altogether! The islets might not be doing very well and this results is not credible.

A: This comment refers to Fig 3F, where we showed that COP1 knockdown in mouse islet diminishes basal insulin secretion and, more strikingly, GSIS. In fact, COP1 knockdown only reduces insulin secretion from WT islets at 2.8mM glucose by ~2 fold, but caused 7 fold reductions at 16.8 mM glucose. Importantly, COP1-KD more pronouncedly affects Het islets (4.5-fold reduction at 2.8mM glucose, 16.7 fold reduction at 16.8mM glucose). These results are consistent with CRL4^{COP1} being more active under high glucose or in Het islets. Also shown in this figure, weak but significant GSIS was observed in COP1-KD islets, confirming that these islets were functional. These results are consistent with the report by Suriben et al (Cell, 2016, 163, 1457), where they showed islets from COP1-KO mice displayed greatly reduced GSIS (Fig 3D of their paper).

5) To refer to incubation of cells at different glucose (p. 14, lines 2. as 'feeding' is stretching it a bit....

A: We are thankful to reviewer 5 for this feedback. Identifying “glucose” as the direct stimulus is in fact more exciting. We have rephrased the sentence as follows: ‘Glucose in medium promotes CRL4^{COP1} complex assembly in βcells (Fig 4F), whereas refeeding fasted mice promotes CRL4^{COP1} complex assembly in pancreas (Fig 5A), suggesting nutrient regulation of a CRL4^{COP1}-mediated degradation pathway’ (page 14, line 8-11, highlighted in red). To be consistent, we have also changed the labels in Fig 4F and 4G from “fast/refed” to “glucose+/-/rescue”.

6) Page 15. Human islet data. How many human islet preparations were used? It is good practice to present donor characteristics in a supplementary table (sex, bmi, age)

A: Three human islet preparations were used for Fig 6A-D. We hoped to provide donor characteristics at the beginning. However, such information were collected by dedicated Organ Procurement Organization in a double-blinded manner, as such, our clinician collaborators had no access to these details. Nonetheless, the donors are metabolically healthy for transplantation surgery, as evaluated by the OPO. We have edited the corresponding method section, emphasizing on the donors being metabolically healthy.

7) If blood glucose concentrations in Fig. 1F-G (and elsewhere) are to be quoted as % (of t=0), then the legend should contain the information on how much 100% is! The effect on insulin tolerance in Fig. 1F is best described as 'marginal'.

In multiple papers that we referenced while carrying out this study, ITT data are all presented as “glucose%”, without specifying how much 100% is (eg: Cell, 2015, 163:1457-1467, PMID: 26627735, Fig 1I; Cell, 2010, 143:897-910, PMID: 21145457, Fig 3B, 5D; PLOS Biology, 2013, 11: e1001541, PMID: 23630453, Fig 4H). Presenting ITT data as “glucose%” is a common practice, since each individual

mouse has its own and distinct glucose level at t=0 (4-6 h fasting), meaning that the 100% reference is different for every single mouse, even in the same treatment group. Nevertheless, the average values of blood glucose levels at t=0 for all the ITT experiments are presented below. These values have now been added into the corresponding figure legends, highlighted in red.

For Fig 1F (2.5 month ITT), 100% means on average 6.4 ± 0.5 mM blood glucose for WT, and 7.3 ± 0.8 mM blood glucose for Het.

For Fig 1G (ITT of streptozocin-treated mice), 100% means on average 14.0 ± 5.4 mM blood glucose for WT, and 11.2 ± 2.2 mM blood glucose for Het.

For Fig S1D (4 month ITT), 100% means on average 9.9 ± 1.3 mM blood glucose for WT, and 8.3 ± 1.1 mM blood glucose for Het.

For Fig S1E (ITT before and after Metformin treatment), 100% means on average 8.8 ± 0.7 mM blood glucose for WT without Metformin treatment, 8.4 ± 0.8 mM for WT with Metformin treatment, 9.2 ± 0.9 mM for Het without Metformin treatment, and 7.6 ± 0.9 mM for Het with Metformin treatment.

For Fig S5C (ITT for NCD/HFD mice with/without MLN treatment) , 100% means on average 7.3 ± 0.6 mM blood glucose for NCD without MLN, 7.5 ± 0.5 mM for HFD with MLN, 9.8 ± 1.5 mM blood glucose for HFD without MLN, and 7.9 ± 0.6 mM for HFD with MLN.

For Fig S5F (ITT before ob/ob mice with/without MLN treatment), 100% means on average 16.3 ± 2.3 mM blood glucose for ob/ob mice without MLN treatment, and 12.2 ± 1.5 mM blood glucose for ob/ob mice with MLN treatment.

Regarding the insulin tolerance data in Fig. 1F, we concur with this reviewer that this effect is not very striking, although it is statistically significant ($p < 0.01$, two way ANOVA). Therefore, we have described this data more moderately on page 7 line 10.

8) Fig. 1J: y-axis (normalised to basal?)

A: We have updated the y-axis of Fig 1J as Relative insulin secretion, normalized to basal.

9) Fig. 3A. Were islets pretreated for 2h with MLN or were experiments conducted in the continued presence of the compound? The effects are remarkably strong (>90%) even in WT islets and yet there is no difference on serum insulin? There is a glaring discrepancy between the data in fig. 3A-B! I would feel more comfortable if the authors could show that MLN is not simply poisoning the islets! (what happens if MLN is applied in the calcium measurements?)

Fig. 3A described experiments whereby mouse islets were pretreated for 2h with MLN (islets were left to balance at 2.8 mM glucose buffer during this 2h time, as described in the methods). During subsequent measurement of insulin secretion, MLN was also added to the buffer. As shown in Fig 3A, MLN reduces basal insulin secretion from WT islets by just 2-fold, but inhibits GSIS by 10-fold, consistent with an important role of CRL disinhibition during GSIS. In vivo, MLN clearly affects serum insulin in Het (Fig 3B), HFD (Fig 7C), and ob/ob mice (Fig 7H). These are mice that would display more active CRL4^{COP1} due to genetic reasons (Het) or

hyperglycemia (HFD and ob/ob). By comparison, MLN does not significantly reduce serum insulin in WT mice, both when used as a control for Het mice (Fig 3B) and for HFD mice (Fig 7C). So the data consistently suggest that MLN has relative modest effect under low glucose, but is very effective in curbing pathological GSIS.

We do not think that MLN is simply poisoning the islets for the follow reasons: First, as suggested by this reviewer, we have examined the effect of MLN treatment on depolarization-induced Ca^{2+} influx. No statistically significant difference was seen, indicating that MLN had no effect on events up to the exocytosis-triggering signal. This new data is presented in Fig S2B, and is described on page 9, highlighted in red. Second, as shown in Fig 3A, even in the presence of MLN, insulin secretion is significantly higher for the high-glucose groups (7+8) than the low glucose groups (5+6), suggesting that MLN-treated islets still retain weaker but highly significant GSIS, as such, they are functional islets. To emphasize these points, we have edited the relevant content on page 9, starting line 19 (highlighted in red).

10) Statistical analyses. The authors clearly like Student's t-test and use it widely - also when it may not be appropriate. Given that many of the effects they report are relatively small, it is important that the appropriate statistical tests are used to bolster their conclusions.

A: We might not be clear in the statistical methods used in individual experiments, and apologize for the misunderstanding. We used Student's t-test for pairwise comparison with two experimental datasets (e.g. shCtrl vs shGeneX #1; shCtrl vs shGeneX #2). We have now put a line on top of the bar graph to more clearly indicate the two samples chosen for T-test. For figures comparing two sets of samples with Time as the 3rd variant, such as Figs 1A, 1E-G, 7B, 7F, we used two-way ANOVA. We consulted with our Statistics department colleagues, and confirmed that the statistical analyses we chose are appropriate.

We would like to thank you and the reviewers for the time and invaluable insights, which help to significantly improve and strengthen our current work. We hope that you will find this letter and the edited manuscript address the reviewers' concerns.

On behalf of all authors,

Feng Rao

Reviewers' Comments:

Reviewer #5:

Remarks to the Author:

I compliment the authors for their diligence dealing with my comments.

I believe the authors have adequately addressed most of my concerns.

It is good that they now define '100%'. I am sure that the authors agree that because poor practices have been used previously (and got published in high-profile journals), this is not an excuse!

Although I am not a statistician, I remain unpersuaded that the authors' extensive use of Student's t-test is always appropriate.

Reviewer #6:

Remarks to the Author:

I was asked to provide a statistical review of this manuscript. Here is what I found.

(1) In the Statistics section of Nature-research's Reporting Summary checklist, the authors checked the "confirmed" checkbox for the exact sample size (n) for each experimental group or condition. But in fact, the exact n per group is almost always missing. This is the opposite of what the authors say when they check the "confirmed" checkbox. Only Figure 1a shows the sample size per group, and it is 10 per group. However, Figures 1b through 1k have different n's per group. Sometimes we can count in each figure how many per group, but sometimes we cannot.

Somewhere in the figure legend for Figure 1, the authors need to tell the reader how many mice there are in each group, and they need to do this for each and every panel in the figure. In addition, the authors need to do the same with the figure legends for Figures 2 through 8, particularly when the number per group changes between panels inside the same figure (which it frequently does).

(2) The authors also checked the "confirmed" checkbox for a statement on whether measurements were taken from distinct samples or whether the same sample was measured repeatedly. But in fact, the authors make no such statement anywhere in the manuscript. This is important because, on Pages 23 and 24 of the manuscript, for the Glucose Tolerance Test, the Insulin Tolerance Test, and the Glucose Stimulated Insulin Secretion Assay, each paragraph says that blood was collected at many time points to measure glucose levels or insulin levels. So here is my question about the Glucose Tolerance Test. When blood was collected at 0, 5, 10, 15, 30, 45, 60, 90, and 120 mins, was the blood collected longitudinally (i.e., repeatedly) from the same mouse at all of those time points? Or did the authors have to sacrifice one mouse per point to get enough blood per time point? I have the same question about the Insulin Tolerance Test and about the Glucose Stimulated Insulin Secretion Assay. If any of these assays involved collecting the blood longitudinally (i.e., repeatedly) from the same mouse at multiple time points, then the authors need to say this in their Methods. Conversely, if any of these assays involved sacrificing a mouse at each time point to get enough blood, then the authors need to say this instead.

(3) Error Bars in the Plots. Clearly defined error bars are present, but what they represent is absent from the figure legends. Based on how large they are, sometimes the error bars appear to represent either standard deviations (SDs) or 95% confidence intervals (CIs), but sometimes they appear instead to represent standard errors (SEs). Each figure with figure legend needs to stand on its own, and not rely on the manuscript's text for critical information. Therefore, each figure legend needs to tell the reader whether the error bars represent SDs, SEs, or CIs. Currently, the figure legends do not do this.

(4) In the statistical analysis paragraph, the first sentence says that all results are presented as the mean and standard error of at least three independent experiments with individual data points shown in the plots when possible. But this statement is clearly wrong if you look at the plots.

Wherever the plots show the individual datapoints, the datapoints represent different mice within the same experiment, not separate independent experiments.

(5) In the statistical analysis paragraph, the second sentence says that statistical significance was calculated by two-tailed Student's t-test or two-way repeated-measures ANOVA (* $p < 0.05$, ** $p < 0.01$, *** $p < 0.001$). This sentence is too vague and too lacking in detail, particularly about the two-way repeated-measures ANOVA. The "two-way" part means there are two factors involved. What are the two factors? Which factor is the within-subjects factor and which factor is the between-subjects factor? Did the authors do any ANOVA post-hoc comparisons between genotype means at each time point? Or did the authors test only the Genotype main effect, the Time Point main effect, and the Genotype-x-Timepoint Interaction from the ANOVA model? My impression is that the authors use Student's T-tests instead of ANOVA post-hoc comparison machinery to compare genotype means at each time point, but the authors do not provide any detail on this. Which reinforces my initial point about how this sentence is too vague and too lacking in detail regarding how the two-way repeated-measures ANOVA was conducted.

(6) In Figure 1a and Figure 1f, where the separation between curves within each figure is annotated with a "***" denoting $p < 0.01$, does the $p < 0.01$ refer just to the genotype difference at the final time point? Or does it refer instead to the Genotype main effect? Or does it refer instead to the Genotype-x-Timepoint interaction? These details are missing.

(7) In the figure legends, whenever the authors provide a p-value from what they say is "two way ANOVA", they do not say that it's repeated-measures ANOVA, and they should. They also do not say what is being marked as significantly different between the curves, and they should. Does the significance refer only to the genotype difference at the final time point? Does the significance refer instead to the overall separation between each genotype's curves (the Genotype main effect)? Or does the significance refer to how the curves fall together at early time points only to diverge at later time points (the Genotype-x-Timepoint interaction)? Clearly some clarification among these different possibilities is needed whenever the authors present a p-value from a repeated-measures ANOVA.

That is the end of my statistical review of this manuscript.

Reviewer #6 (Remarks to the Author):

I was asked to provide a statistical review of this manuscript. Here is what I found.

(1) In the Statistics section of Nature-research's Reporting Summary checklist, the authors checked the "confirmed" checkbox for the exact sample size (n) for each experimental group or condition. But in fact, the exact n per group is almost always missing. This is the opposite of what the authors say when they check the "confirmed" checkbox. Only Figure 1a shows the sample size per group, and it is 10 per group. However, Figures 1b through 1k have different n's per group. Sometimes we can count in each figure how many per group, but sometimes we cannot. Somewhere in the figure legend for Figure 1, the authors need to tell the reader how many mice there are in each group, and they need to do this for each and every panel in the figure. In addition, the authors need to do the same with the figure legends for Figures 2 through 8, particularly when the number per group changes between panels inside the same figure (which it frequently does).

A: We appreciate this comment on clarifying sample size (n). We did not initially provide sample size information, since sample sizes are presented as the number of dots in bar graphs. Sample size (n) for scatter plots can also be found from co-submitted supplementary data. We have now included sample size (n) for all figures.

(2) The authors also checked the "confirmed" checkbox for a statement on whether measurements were taken from distinct samples or whether the same sample was measured repeatedly. But in fact, the authors make no such statement anywhere in the manuscript. This is important because, on Pages 23 and 24 of the manuscript, for the Glucose Tolerance Test, the Insulin Tolerance Test, and the Glucose Stimulated Insulin Secretion Assay, each paragraph says that blood was collected at many time points to measure glucose levels or insulin levels. So here is my question about the Glucose Tolerance Test. When blood was collected at 0, 5, 10, 15, 30, 45, 60, 90, and 120 mins, was the blood collected longitudinally (i.e., repeatedly) from the same mouse at all of those time points? Or did the authors have to sacrifice one mouse per point to get enough blood per time point? I have the same question about the Insulin Tolerance Test and about the Glucose Stimulated Insulin Secretion Assay. If any of these assays involved collecting the blood longitudinally (i.e., repeatedly) from the same mouse at multiple time points, then the authors need to say this in their Methods. Conversely, if any of these assays involved sacrificing a mouse at each time point to get enough blood, then the authors need to say this instead.

A: We thank the reviewer for this comment. For experiments involving time as the 3rd variant, such as The GTT, ITT and GSIS assays, experiments were performed longitudinally, whereby blood were collected from the same mouse at different time points. For experiments without involving time as the 3rd variant, such as blood glucose, serum insulin, plasma c-peptide, relative islet insulin secretion, and other measurements, samples were collected from distinct mice at one time.

We have now edited page 23 and 24 of the manuscript to clarify this difference, with the changes highlighted in red.

(3) Error Bars in the Plots. Clearly defined error bars are present, but what they represent is absent from the figure legends. Based on how large they are, sometimes the error bars appear to represent either standard deviations (SDs) or 95% confidence intervals (CIs), but sometimes they appear instead to represent standard errors (SEs). Each figure with figure legend needs to stand on its own, and not rely on the manuscript's text for critical information. Therefore, each figure legend needs to tell the reader whether the error bars represent SDs, SEs, or CIs. Currently, the figure legends do not do this.

A: We concur with this reviewer that error bars needs to be clearly defined, although they can be figured out with the data points presented in the graph. We have now added in each figure legend a sentence clearly stating “Data are presented as mean \pm SEM/SD”. These updates are highlighted in red.

(4) In the statistical analysis paragraph, the first sentence says that all results are presented as the mean and standard error of at least three independent experiments with individual data points shown in the plots when possible. But this statement is clearly wrong if you look at the plots. Wherever the plots show the individual datapoints, the datapoints represent different mice within the same experiment, not separate independent experiments.

A: To more accurately reflect how the experiments were performed, this sentence (page 28, highlighted in red) has been rephrased as: “All results are presented as the mean with standard error (SEM) or standard deviation (SD) of at least triplicates. The sample size (n) defines the number of mice for animal experiments and the number of replicates for cell studies. All data-points are displayed in the plots when possible.”

(5) In the statistical analysis paragraph, the second sentence says that statistical significance was calculated by two-tailed Student's t-test or two-way repeated-measures ANOVA (*p<0.05, **p<0.01, ***p<0.001). This sentence is too vague and too lacking in detail, particularly about the two-way repeated-measures ANOVA. The "two-way" part means there are two factors involved. What are the two factors? Which factor is the within-subjects factor and which factor is the between-subjects factor? Did the authors do any ANOVA post-hoc comparisons between genotype means at each time point? Or did the authors test only the Genotype main effect, the Time Point main effect, and the Genotype-x-Timepoint

Interaction from the ANOVA model? My impression is that the authors use Student's T-tests instead of ANOVA post-hoc comparison machinery to compare genotype means at each time point, but the authors do not provide any detail on this. Which reinforces my initial point about how this sentence is too vague and too lacking in detail regarding how the two-way repeated-measures ANOVA was conducted.

A: For experiments without involving time as the 3rd variant, we used two-tailed student's test from EXCEL for statistical analysis.

For experiments involving time as the 3rd variant, we analyzed the data using ordinary two-way ANOVA test package from the software Sigmaplot (version 12.3) for statistical analysis. The repeated-measure ANOVA is less appropriate, since our sample size "n" refers to the number of mice and not repeated-measure of the same mice. In the current two-way ANOVA test, factor A is time, and factor B is genotype. No between-subjects factor was defined. Only the genotype main effect was tested to support our claim of genetic effects. Genotype-x-Timepoint Interaction or post-hoc comparisons between genotype means at each time point were not analyzed.

We have now added these details to the statistical analysis paragraph, highlighted in red.

(6) In Figure 1a and Figure 1f, where the separation between curves within each figure is annotated with a "***" denoting $p < 0.01$, does the $p < 0.01$ refer just to the genotype difference at the final time point? Or does it refer instead to the Genotype main effect? Or does it refer instead to the Genotype-x-Timepoint interaction? These details are missing.

A: The p values in Fig1a and Fig1f refer to Genotype main effect, not genotype difference at the final time point, nor Genotype-x-Timepoint interaction. This has now been added to the corresponding figure legends, highlighted in red.

(7) In the figure legends, whenever the authors provide a p-value from what they say is "two way ANOVA", they do not say that it's repeated-measures ANOVA, and they should. They also do not say what is being marked as significantly different between the curves, and they should. Does the significance refer only to the genotype difference at the final time point? Does the significance refer instead to the overall separation between each genotype's curves (the Genotype main effect)? Or does the significance refer to how the curves fall together at early time points only to diverge at later time points (the Genotype-x-Timepoint interaction)? Clearly some clarification among these different possibilities is needed whenever the authors present a p-value from a repeated-measures ANOVA.

A: As explained in our reponse to point 5. We used ordinary two way ANOVA , not repeated ANOVA. The significance refers to the overall separation between each genotype's curves, i.e., the Genotype main effect). This information has been added to the methods paragraph on page 28, highlighted in red. We have also defined what is being compared and tested for the p-values from all two way ANOVA analysis in the corresponding figure legends, highlighted in red.

Reviewers' Comments:

Reviewer #6:

Remarks to the Author:

The authors responded adequately to all of my statistical concerns, but now I have a new concern, and a new request.

First, I want to compliment the authors for how meticulously they addressed my concerns. Because they were so thorough, I think that the manuscript is now much more transparent than it was, and much improved as a result.

Now for my new concern. The authors say that, for experiments involving time as the 3rd variant, they analyzed the data using ordinary two-way ANOVA test package using Sigmaplot. And they also say that the repeated-measure ANOVA is less appropriate, because their sample size "n" represents the number of mice and not the number of repeated measures on the same mouse. That answer makes me think that the authors were looking at a 1-way repeated-measures ANOVA package in Sigmaplot. That is the wrong package. What the authors want to look for instead is the 2-way repeated-measures ANOVA package in Sigmaplot. It is different from the ordinary 2-way ANOVA package in Sigmaplot.

More generally, in any experiment where something is measured repeatedly in the same mouse over time, the different measurements from the same mouse are correlated with each other, and are not independent. This is called autocorrelation. The ordinary 2-way ANOVA treats all of the data as independent, which means that the ordinary 2-way ANOVA wrongly ignores the autocorrelation. On the other hand, the 2-way repeated-measures ANOVA takes the autocorrelation into account when it calculates p-values. For this reason, when something is measured repeatedly in the same mouse over time, it is more appropriate to use the 2-way repeated-measures ANOVA, and less appropriate to use the ordinary 2-way ANOVA.

Therefore, I have a request. For every experiment that involved time as the 3rd variant, such as the GTT, ITT, and GSIS assays, as well as body weight, I request that the authors re-analyze their experiment's data using the 2-way Repeated Measures ANOVA package in Sigmaplot. The authors should not use ordinary 2-way ANOVA for these experiments. The ordinary 2-way ANOVA is giving the authors invalid P-values. The 2-way repeated-measures ANOVA will give the authors valid ones.

To help the authors out, I found a YouTube video that shows how to do 2-way repeated-measures ANOVA in Sigmaplot. The example that the video shows is for "Two Way Repeated Measures ANOVA (One Factor Repetition)". That is exactly the kind of 2-way repeated-measures ANOVA that the authors will need for their re-analysis. The YouTube video is located at:
<https://www.youtube.com/watch?v=GatgXQh4w-8>

I wish the authors success in their re-analysis.

Thank you for your continued efforts in improving our manuscript, and for the opportunity to revise MS #: 20-13724C. We are glad to be able to address most of the concerns of Reviewer 6. The new comment/request from reviewer #6 also helped to improve the manuscript. We have therefore revised the manuscript accordingly.

Reviewer #6 (Remarks to the Author):

The authors responded adequately to all of my statistical concerns, but now I have a new concern, and a new request.

First, I want to compliment the authors for how meticulously they addressed my concerns. Because they were so thorough, I think that the manuscript is now much more transparent than it was, and much improved as a result.

A: We are thankful to this expert reviewer for the excellent comments that has been improving the manuscript.

Now for my new concern. The authors say that, for experiments involving time as the 3rd variant, they analyzed the data using ordinary two-way ANOVA test package using Sigmaplot. And they also say that the repeated-measure ANOVA is less appropriate, because their sample size "n" represents the number of mice and not the number of repeated measures on the same mouse. That answer makes me think that the authors were looking at a 1-way repeated-measures ANOVA package in Sigmaplot. That is the wrong package. What the authors want to look for instead is the 2-way repeated-measures ANOVA package in Sigmaplot. It is different from the ordinary 2-way ANOVA package in Sigmaplot.

A: We appreciate this concern from the reviewer. We initially choose ordinary two-way ANOVA because the advice we get from a statistics academic was that "repeated measures" is for experiments where one measures the same data point again and again, which was not case for our study. We then followed the following Youtube viedo to perform the ordinary two-way ANOVA analysis in Sigmaplot:

https://www.youtube.com/watch?v=WLeSu3kPEUw&ab_channel=DoryVideo

We now see the point of auto-correlation have now performed 2-way repeated-measures ANOVA (see below).

More generally, in any experiment where something is measured repeatedly in the same mouse over time, the different measurements from the same mouse are correlated with each other, and are not independent. This is called autocorrelation. The ordinary 2-way ANOVA treats all of the data as independent, which means that the ordinary 2-way ANOVA wrongly ignores the autocorrelation. On the other hand, the 2-way repeated-measures ANOVA takes the autocorrelation into account when it calculates p-values. For this reason, when something is measured repeatedly in the same mouse over time, it is more appropriate to use the 2-way repeated-measures ANOVA, and less appropriate to use the ordinary 2-way ANOVA.

Therefore, I have a request. For every experiment that involved time as the 3rd variant, such as the GTT, ITT, and GSIS assays, as well as body weight, I request that the authors re-analyze their experiment's data using the 2-way Repeated Measures ANOVA package in Sigmaplot. The authors should not use ordinary 2-way ANOVA for these experiments. The ordinary 2-way ANOVA is giving the authors invalid P-values. The 2-way repeated-measures ANOVA will give the authors valid ones.

To help the authors out, I found a YouTube video that shows how to do 2-way repeated-measures ANOVA in Sigmaplot. The example that the video shows is for "Two Way Repeated Measures ANOVA (One Factor Repetition)". That is exactly the kind of 2-way repeated-measures ANOVA that the authors will need for their re-analysis. The YouTube video is located at:

<https://www.youtube.com/watch?v=GatgXQh4w-8>

A: We appreciate the patient explanation from this reviewer and concur with this reviewer that data from different time points of the same mice are not independent indeed. We have therefore revised the manuscript by performing two-way repeated measures ANOVA test for experiments involving time as the 3rd variant that were previously analyzed using ordinary two-way ANOVA, namely Figs 1a, 1f, 1g, 3b-c, 7b, 7f, s1, s1d, s1e, s1g, and s5b-f. Since these experiments examine the effect of a combination of time and genotype/treatment, the "genotype-x-time interaction" or "treatment-x-time interact" were tested and p-values are reported. Corresponding changes made to the text in the Method and Figure legend sections are highlighted in red.

To facilitate the evaluation of this revision, we've also attached the new data analysis sigmaplot files for Fig 1a and Fig 3b. Fig 1a exemplifies a "Genotype (factor A) x time (factor B)" plot, and Fig 3b exemplifies "Treatment (factor A) x time (factor B)" plot. They represent the two types of experiments that involve time as the 3rd variant, but was previously analyzed by ordinary two-way ANOVA.

We would like to thank reviewer 6 again for the effort to help improve our statistical analysis. We hope that you will find this letter and the edited manuscript address this reviewer's concerns.

On behalf of all authors,

Feng Rao

Reviewers' Comments:

Reviewer #6:

Remarks to the Author:

The authors have adequately addressed my last concern. I have no new concerns to express. If the editors decide to approve the manuscript for publication, then I support the editors' decision.